# The AMPK agonist 5-aminoimidazole-4-carboxamide ribonucleotide (AICAR), but not metformin, prevents inflammation-associated cachectic muscle wasting

Derek T Hall[1,2], Takla Griss[2,3], Jennifer F Ma[1,2], Brenda Janice Sanchez[1,2], Jason Sadek[1,2], Anne Marie K Tremblay[1,2], Souad Mubaid[1,2], Amr Omer[1,2], Rebecca J Ford[4], Nathalie Bedard[5], Arnim Pause[1,2], Simon S Wing[5], Sergio Di Marco[1,2], Gregory R Steinberg[4], Russell G Jones[2,3] & Imed-Eddine Gallouzi[1,2,6,*] (ID)

## Abstract

Activation of AMPK has been associated with pro-atrophic signaling in muscle. However, AMPK also has anti-inflammatory effects, suggesting that in cachexia, a syndrome of inflammatory-driven muscle wasting, AMPK activation could be beneficial. Here we show that the AMPK agonist AICAR suppresses IFNγ/TNFα-induced atrophy, while the mitochondrial inhibitor metformin does not. IFNγ/TNFα impair mitochondrial oxidative respiration in myotubes and promote a metabolic shift to aerobic glycolysis, similarly to metformin. In contrast, AICAR partially restored metabolic function. The effects of AICAR were prevented by the AMPK inhibitor Compound C and were reproduced with A-769662, a specific AMPK activator. AICAR and A-769662 co-treatment was found to be synergistic, suggesting that the anti-cachectic effects of these drugs are mediated through AMPK activation. AICAR spared muscle mass in mouse models of cancer and LPS induced atrophy. Together, our findings suggest a dual function for AMPK during inflammation-driven atrophy, wherein it can play a protective role when activated exogenously early in disease progression, but may contribute to anabolic suppression and atrophy when activated later through mitochondrial dysfunction and subsequent metabolic stress.

**Keywords** AMPK; cachexia; inflammation; iNOS; metabolism
**Subject Categories** Cancer; Immunology; Metabolism

## Introduction

Cachexia is a wasting syndrome that often occurs as a comorbidity with chronic pro-inflammatory diseases, such as cancer, HIV infection, and sepsis (Fearon *et al*, 2011; Blum *et al*, 2014). Cachexia is primarily characterized by a progressive and extensive loss of skeletal muscle mass and strength, but can also present with loss of fat mass, anorexia, and cardiac atrophy and remodeling (Fearon *et al*, 2012; Groarke *et al*, 2013). The prevalence of cachexia in patients varies depending on the type of disease. For example, it is estimated that approximately half of all cancer patients experience cachexia (von Haehling & Anker, 2010, 2014). It is well established that onset of cachexia negatively impacts disease outcome, reducing the effectiveness of primary disease treatment and increasing patient morbidity and mortality (Andreyev *et al*, 1998; Prado *et al*, 2007; Utech *et al*, 2012; Vaughan *et al*, 2013; Vigano *et al*, 2017).

While there are numerous symptoms of the cachectic state, one of the more debilitating aspects of this condition is the dramatic loss of skeletal muscle tissue (Fearon *et al*, 2011). Although the mechanisms behind cachectic muscle wasting are complex, it is believed that one of the primary triggers of muscle atrophy is the chronic elevation of pro-inflammatory cytokines (e.g., IL-1, IL-6, TNFα, IFNγ) (Morley *et al*, 2006; Argiles *et al*, 2009; Tisdale, 2009; Fearon *et al*, 2012). In keeping with this, induction of muscle atrophy can be recapitulated in culture and *in vivo* by exposure to different cytokine combinations [e.g., IFNγ and TNFα (Guttridge *et al*, 2000; Acharyya *et al*, 2004; Di Marco *et al*, 2005, 2012) or IL-6 (Bonetto *et al*, 2012; White *et al*, 2012)]. Extended cytokine exposure results in the continued activation of inflammatory signaling within muscle cells, leading to the expression of pro-cachectic genes (Guttridge *et al*, 2000; Bonetto *et al*, 2011; Hall *et al*, 2011; Fearon *et al*, 2012; Bonaldo & Sandri, 2013). For example, we and others have

1 Department of Biochemistry, McGill University, Montreal, QC, Canada
2 Rosalind and Morris Goodman Cancer Centre, Montreal, QC, Canada
3 Department of Physiology, McGill University, Montreal, QC, Canada
4 Division of Endocrinology and Metabolism, Department of Medicine, McMaster University, Hamilton, ON, Canada
5 Department of Medicine, McGill University and the Research Institute of the McGill University Health Centre, Montreal, QC, Canada
6 Life Sciences Division, College of Sciences and Engineering, Hamad Bin Khalifa University (HBKU), Doha, Qatar
*Corresponding author. Tel: +1 514 398 4537; E-mails: imed.gallouzi@mcgill.ca; igallouzi@qf.org.qa

demonstrated that inducible nitric oxide synthase (iNOS) is dramatically upregulated during cytokine-driven muscle wasting and that the production of reactive nitrogen compounds, such as nitric oxide (NO), by this enzyme contributes to the pathogenesis of cachexia (Buck & Chojkier, 1996; Di Marco et al, 2005, 2012; Ramamoorthy et al, 2009; Hall et al, 2011). Recent evidence suggests that cytokine exposure also alters the metabolism of muscle. Indeed, it has been shown in several models of cachexia and inflammation-driven wasting that muscle undergoes a Warburg-like increase in aerobic glycolysis and mitochondrial abnormalities (Barreiro et al, 2005; Julienne et al, 2012; White et al, 2012; Der-Torossian et al, 2013; Fontes-Oliveira et al, 2013; McLean et al, 2014).

The metabolic regulating enzyme, AMP-activated protein kinase (AMPK), has been associated with cytokine- and cancer-driven muscle wasting (White et al, 2011, 2013). AMPK is a heterotrimeric complex (composed of a catalytic α-subunit, linker β-subunit, and regulatory γ-subunit) that responds to cellular energy levels (Hardie et al, 2012). Activation of AMPK has been shown to suppress anabolic signaling through mTOR and has been found to increase the expression of muscle-specific E3-ligases (Krawiec et al, 2007; Nakashima & Yakabe, 2007; Shaw, 2009). However, AMPK has also been shown to have potent anti-inflammatory effects in a variety of cell types (Galic et al, 2011; Salminen et al, 2011; Mounier et al, 2013; O'Neill & Hardie, 2013). The anti-inflammatory function suggests that AMPK activation could be beneficial for muscle atrophy induced by chronic inflammation. Consistent with this concept, genetic deletion of skeletal muscle AMPK leads to the acceleration of aging-induced sarcopenia (Bujak et al, 2015). Further, an AMPK stabilizing peptide was recently shown to be effective at preventing adipose tissue wasting in cancer cachexia (Rohm et al, 2016). Therefore, there is an apparent contradiction for the role of AMPK during cachectic muscle wasting: While its association with atrophic signaling suggests AMPK can contribute to muscle wasting during cachexia, the anti-inflammatory functions of AMPK suggest that it could also prevent cytokine-driven atrophy.

Here, we tested the hypothesis that compounds that activate AMPK could prevent cytokine-driven muscle wasting. To do so, we assessed the impact of two well-known AMPK activators—AICAR and metformin—on atrophy in cultured myotubes treated with the pro-inflammatory cytokines IFNγ and TNFα (Towler & Hardie, 2007; Viollet et al, 2012). These compounds activate AMPK through distinct mechanisms. AICAR is phosphorylated by cellular kinases to form ZMP, which acts as an AMP mimetic, binding directly to and activating AMPK (Towler & Hardie, 2007). In contrast, the biguanide metformin inhibits Complex I of the electron transport chain, leading to an indirect activation of AMPK by increasing cytoplasmic AMP levels (Viollet et al, 2012). Surprisingly, we found that while AICAR, metformin, and IFNγ/TNFα treatment activated AMPK, only AICAR prevented IFNγ/TNFα-induced atrophy. In addition, AICAR, but not metformin, was found to partially restore normal metabolic function and inhibit the pro-cachectic iNOS/NO pathway. The effects of AICAR were blocked by co-treatment with the AMPK inhibitor Compound C and recapitulated with the more specific AMPK activator A-769662 (Cool et al, 2006; Goransson et al, 2007). In addition, A-769662 and AICAR were found to synergistically prevent wasting, suggesting that the effects of these compounds are through AMPK. Finally, AICAR was able to restore muscle mass in multiple murine models of cachectic muscle wasting.

# Results

## Activation of AMPK by AICAR but not metformin prevents muscle wasting

To assess whether AMPK activation could prevent cytokine-induced muscle atrophy, we performed studies in C2C12 myotubes treated with IFNγ and TNFα. These pro-cachectic cytokines are a well-established model to induce a muscle wasting-like phenotype in vitro that begins with signaling events occurring within the first 24 h, followed by atrophy detectable by 48 h and culminating in loss of integrity by 72 h (Di Marco et al, 2005, 2012). To activate AMPK, we used two AMPK activators, AICAR and metformin. As expected, both AICAR and metformin showed increased AMPK phosphorylation at Thr172 (pAMPK), a post-translational modification that is required for AMPK activity, 24 h after treatment (Hardie et al, 2012; Fig 1A). AICAR and metformin treatment also led to the increased phosphorylation of acetyl-CoA carboxylase (ACC) at Ser79 (pACC) (Fig 1A). Acetyl-CoA carboxylase is a well-established downstream target of AMPK and is often used to demonstrate increased AMPK activity within cells (Munday, 2002). Interestingly, IFNγ/TNFα treatment alone also increased pAMPK and pACC levels at 24 h, corroborating previous reports that AMPK phosphorylation increases during the progression of cachexia-induced muscle wasting (Fig 1A; Penna et al, 2010; White et al, 2011, 2013). To further understand the dynamics of AMPK activation in this model, we tested the phosphorylation status of AMPK and ACC over a time course of the first 24 h of cytokine treatment. We observed that while cytokine treatment alone resulted in detectable phosphorylation of ACC at 6–12 h, respectively, co-treatment with AICAR resulted in detectable levels by 30 min (Fig EV1). Metformin treatment, in turn, induced detectable ACC

**Figure 1.  AICAR but not metformin prevents IFNγ/TNFα-induced myotube atrophy.**

A     Western blotting for phospho-Thr172-AMPK (pAMPK), total AMPK, phospho-Ser79-ACC (pACC), and total ACC 24h after treatment. Quantification represents the pAMPK/AMPK and pACC/ACC ratios relative to the non-treated (NT) control.

B     Phase contrast images of fibers 72 h after treatment. Scale bars represent 0.25 mm.

C     Immunofluorescence staining for myoglobin and myosin heavy chain (MyHC) 48 h after treatment. Scale bars represent 50 μm. Quantification represents the average myotube width.

D, E  RT–qPCR analysis of the mRNA levels of MyoD (D) and myogenin (E) 24 h after treatment relative to the NT control.

Data information: All quantifications are of three independent experiments (n = 3) and error bars represent standard error of the mean (SEM). Significance between means was first determined using ANOVA. Significance P-values were calculated using Fisher's LSD. *P < 0.05; **P < 0.01 from NT controls; ††P < 0.01 from IFNγ/TNFα-treated controls.

Source data are available online for this figure.

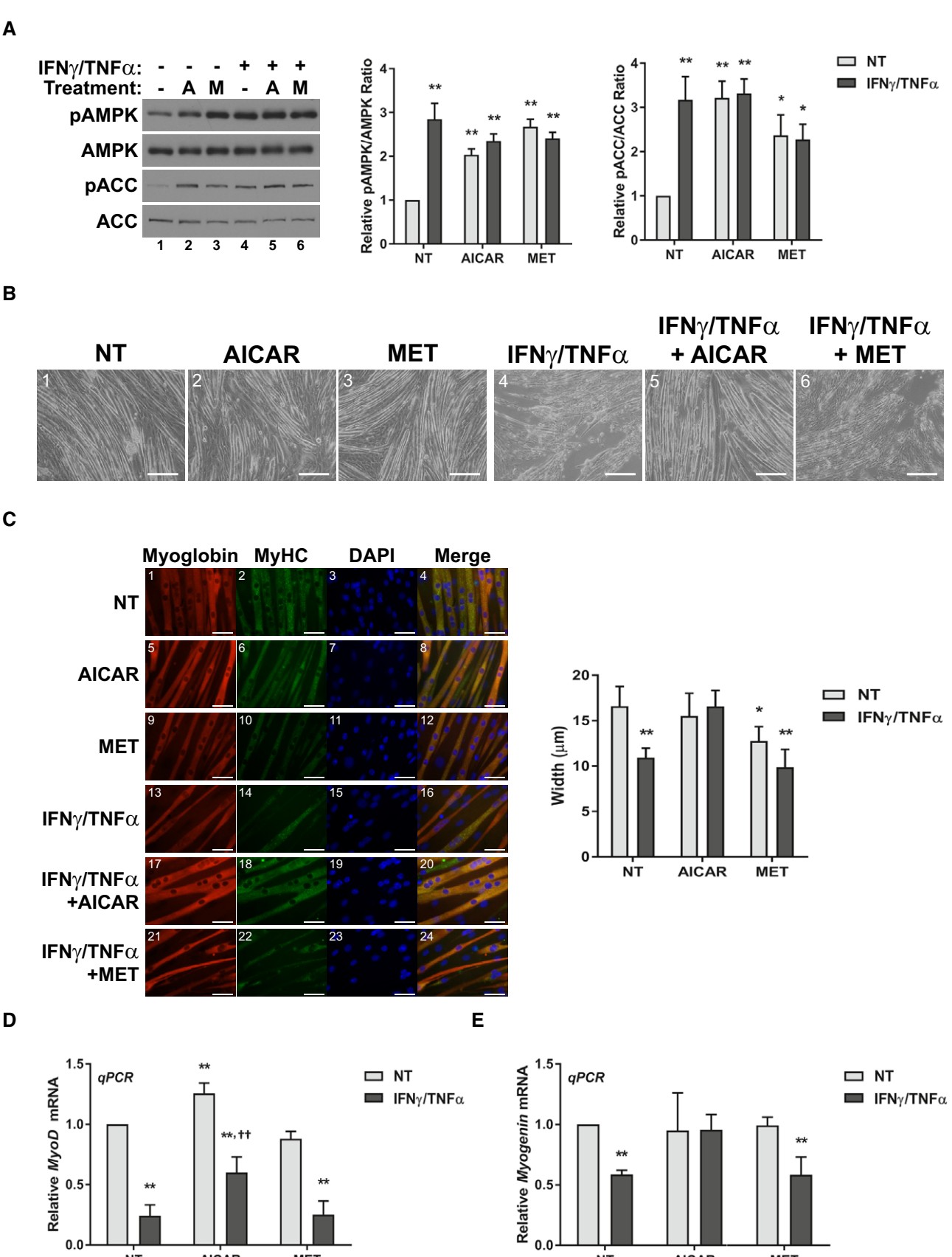

**Figure 1.**

phosphorylation by 1 h (Fig EV1). Thus, while AMPK does seem to be activated by cytokine treatment, eventually, treatment with the AMPK agonists increased AMPK activity at a time when it was not normally induced by cytokine treatment.

Having established the AMPK activation status across the different treatment regimes, we next sought to determine what would be the effect of AMPK agonists on the progression of muscle wasting. Surprisingly, we found that AICAR treatment, but not metformin, was able to prevent the loss of integrity in myotubes exposed to IFNγ/TNFα over a 72-h period (Fig 1B). To determine whether the AICAR treatment also protected myotubes from the initial atrophying that precedes myotube collapse, we measured myotube widths 48 h after cytokine treatment. Again, only AICAR treatment prevented the cytokine-induced atrophy of muscle myotubes (Fig 1C). In contrast, metformin had no effect on cytokine-induced atrophy and showed a trend toward smaller widths when used alone (Fig 1C). Finally, we assessed the mRNA expression levels of the pro-myogenic transcription factors MyoD and myogenin, which are down-regulated within the first 24 h of IFNγ/TNFα treatment (Fig 1D and E). We observed that AICAR treatment, but not metformin, significantly increased the mRNA levels of MyoD in cytokine-treated myotubes compared to cytokine treatment alone, though they were not restored to non-treated control levels (Fig 1D). However, AICAR treatment did restore Myogenin mRNA to basal levels during cytokine treatment (Fig 1E). Taken together, these results clearly demonstrate that, despite both compounds activating AMPK, only co-treatment with AICAR prevents the progression of cytokine-induced myotube wasting.

## AICAR-mediated activation of AMPK restores normal muscle metabolism

In recent years, it has been demonstrated that muscle undergoing cachectic wasting exhibit altered metabolic profiles compared to healthy fibers (Julienne et al, 2012; Der-Torossian et al, 2013; Fontes-Oliveira et al, 2013). Further, given that AMPK is intimately involved in metabolism, we predicted that cytokine treatment would significantly alter metabolic activity in myotube cells, and that the treatment with the AMPK agonists may affect these changes (Hardie et al, 2012). It has been shown that tumor-bearing mice with cachexia exhibit a metabolic signature in muscle characterized by a Warburg-like increase in glycolysis (Der-Torossian et al, 2013). To assess whether this was also the case in our model, we measured the rate of glucose consumption and lactate production, which are indicative of the rate of glycolytic flux. We observed a significant increase in lactate production and glucose consumption in myotubes treated with IFNγ/TNFα, indicative of elevated glycolysis induced by this treatment (Fig 2A and B). Metformin, a known inhibitor of mitochondrial respiration that induces a compensatory increase in glycolysis, also increased lactate production and glucose consumption, as expected, with no additional increase when co-treated with inflammatory cytokines (Fig 2A and B; Viollet et al, 2012). AICAR, on the other hand, did not affect glycolysis on its own and reduced the increase in glycolysis caused by cytokine treatment (Fig 2A and B). Therefore, in keeping with their ability to protect or not against IFNγ/TNFα-induced wasting, AICAR but not metformin was able to prevent the increased glycolytic rate induced by these cytokines.

Changes in glycolytic activity are often, though not always, associated with changes in oxidative respiration in the mitochondria. Cytokine exposure has been associated with impaired mitochondrial function and reduced oxidative capacity. In addition, several reports have found evidence of mitochondrial dysfunction in pre-clinical models of cachexia (Constantinou et al, 2011; Julienne et al, 2012; Tzika et al, 2013). Therefore, we assessed mitochondrial respiration in C2C12 myotubes using the Seahorse XF extracellular flux system. We found that cytokine treatment significantly reduced both the basal and maximal oxygen consumption rates (OCR; Fig 2C). This was associated with a dramatic increase in the extracellular acidification rate (ECAR), in keeping with our findings that cytokine treatment increases glycolytic flux (Fig 2D). Together, the shifts in OCR and ECAR show a dramatic shift in C2C12 myotubes treated with IFNγ/TNFα from an aerobic to glycolytic metabolism (Fig 2E). As expected, metformin, a known mitochondrial inhibitor, induced a similar inhibition of OCR and elevation of ECAR, though not to the same magnitude as cytokine treatment (Fig 2C–E). Metformin co-treatment with IFNγ/TNFα had no additional effects (Fig 2C–E). In contrast, AICAR co-treatment partially restored both basal and maximal respiration and reduced the ECAR (Fig 2C–E). To assess mitochondrial coupling, we compared the respiration rates before and after injection of the ATP-synthase inhibitor oligomycin (Brand & Nicholls, 2011). We found that the decreases in basal respiration during cytokine and metformin treatment were the result of reductions in both coupled and uncoupled respiration (Fig 2F). However, while metformin did not affect the coupling efficiency, IFNγ/TNFα significantly reduced it (Fig 2F). Interestingly, although AICAR co-treatment did not restore basal respiration to its non-treated levels, it did fully recover the coupling efficiency (Fig 2F). Therefore, in cytokine-treated cells co-treated with AICAR, but not metformin, there is a recovery of ATP synthesis-dependent mitochondrial respiration. In contrast, metformin alone impairs mitochondrial respiration and has no recovery effect during cytokine co-treatment. Collectively, the metabolomics analysis shows that cytokines induce a shift toward glycolysis associated with severely impaired mitochondrial oxidative respiration that is blunted by co-treatment with AICAR. In contrast, metformin treatment alone impairs mitochondrial respiration and has no additive effect during co-treatment with cytokines. This suggests that the inability of metformin to recover atrophy during cytokine treatment, unlike AICAR, could be due to a lack of recovery of mitochondrial function.

One potential consequence of altered metabolism is reduced anabolism. Indeed, the inhibition of anabolic signaling is considered to be a key mechanism underlying atrophy in a variety of overlying pro-inflammatory conditions (Rennie et al, 1983; Smith & Tisdale, 1993). Furthermore, AMPK activation has been implicated in suppressing anabolic signaling in cancer cachexia by inhibiting mTOR (White et al, 2013). To assess how AICAR and metformin treatments affect anabolic signaling in cytokine-treated myotubes, we determined the phosphorylation status of the ribosomal protein S6 kinase (S6K) at Thr389 and its target ribosomal protein S6 (S6) at Ser235/236, a downstream target of signaling mTOR (Hornberger et al, 2007; Roux et al, 2007). As expected, cytokine treatment resulted in hypo-phosphorylation of S6K and S6 48 h after treatment, which is indicative of reduced translation initiation (Fig 3A; Roux et al, 2007). AICAR treatment, but not metformin, was able to prevent this decrease

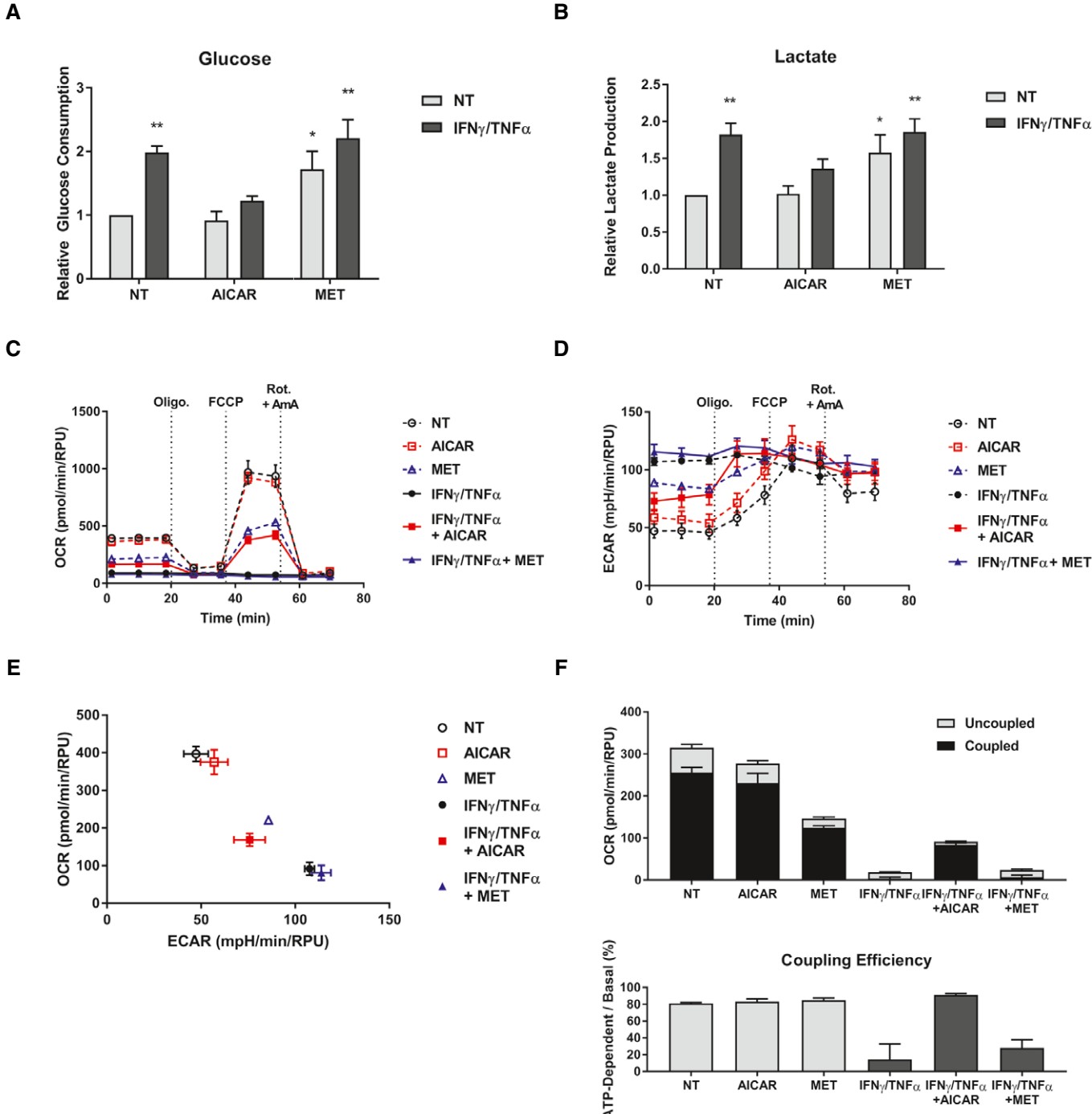

**Figure 2.  AICAR corrects cellular metabolic changes in cytokine-treated myotubes.**

A, B   Rates of glucose consumption (A) and lactate production (B) measured in the media 24 h after treatment relative to the non-treated (NT) control from three independent experiments ($n$ = 3). Error bars represent the SEM.

C–F   Seahorse XF extracellular flux analysis performed on cells 24 h after treatment. Sequential injections of oligomycin (Oligo.), FCCP, and a combination of rotenone and antimycin A (Rot. + AmA) were performed to assess mitochondrial fitness. Flux was normalized to relative protein units (RPU) measured after the run with an SRB assay. Data are representative of three independent experiment ($n$ = 3). Error bars represent the standard deviation of biological replicates (SD). (C) Oxygen consumption rates (OCR). (D) Extracellular acidification rates (ECAR). (E) Coordinate plot of OCR and ECAR showing the cellular metabolic profile. (F) Measurements of uncoupled (oligomycin-resistant) and coupled (oligomycin-sensitive) respiration. Coupling efficiency was calculated as the percentage of basal respiration associated with coupled respiration.

Data information: Significance between means was first determined using ANOVA. Significance $P$-values were calculated using Fisher's LSD. *$P$ < 0.05; **$P$ < 0.01 from NT controls.

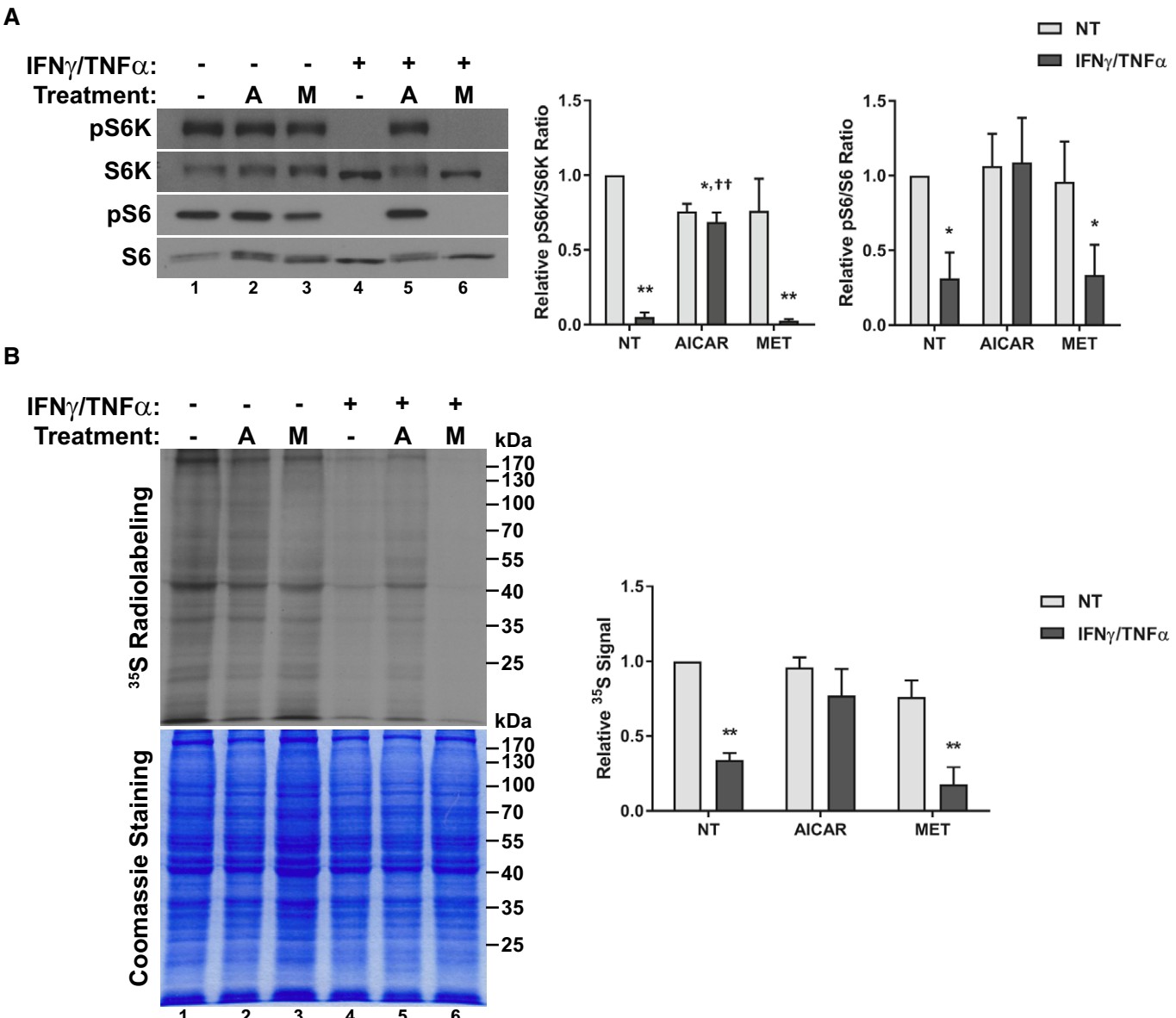

**Figure 3. AICAR restores anabolic signaling and *de novo* protein synthesis in cytokine-treated myotubes.**

A   Western blotting of phospho-Thr389-p70S6K (pS6K), total p70S6K (S6K), phospho-Ser235/236-S6 (pS6), and total S6. Quantification represents the pS6K/S6K ($n = 3$) and pS6/S6 ($n = 4$) ratios relative to the non-treated (NT) control.

B   Radiographic analysis of *de novo* protein synthesis using $^{35}$S-labeling. Quantification represents whole lane radiation signal density standardized to Coomassie staining and relative to NT control levels ($n = 3$).

Data information: Error bars represent the SEM. Significance between means was first determined using ANOVA. Significance *P*-values were calculated using Fisher's LSD. *$P < 0.05$; **$P < 0.01$ from NT controls; ††$P < 0.01$ from IFNγ/TNFα-treated controls.

Source data are available online for this figure.

(Fig 3A). It is important to note that, while previous reports have demonstrated that exogenous activation of AMPK leads to mTOR suppression, the dosage and timing of AICAR and metformin used here had no significant effect on phosphorylation of S6K or S6 on its own (Fig 3A; Williamson *et al*, 2006; Xu *et al*, 2012).

We next sought to directly determine whether these effects on anabolic signaling correlated with changes in general protein biosynthesis. To do so, we performed radio-labeling experiments in

which myotubes treated with IFNγ/TNFα in combination with AICAR or metformin were incubated with L-[$^{35}$S]-methionine and L-[$^{35}$S]-cysteine. Cells treated with IFNγ/TNFα showed a significant decrease in the incorporation of radioactivity, indicating that cytokine treatment suppresses *de novo* protein synthesis (Fig 3B). AICAR treatment, but not metformin, prevented this suppression (Fig 3B). Hence, correlating with a restoration in normal metabolic function, only AICAR appears to be able to restore anabolic signaling in wasting myotubes.

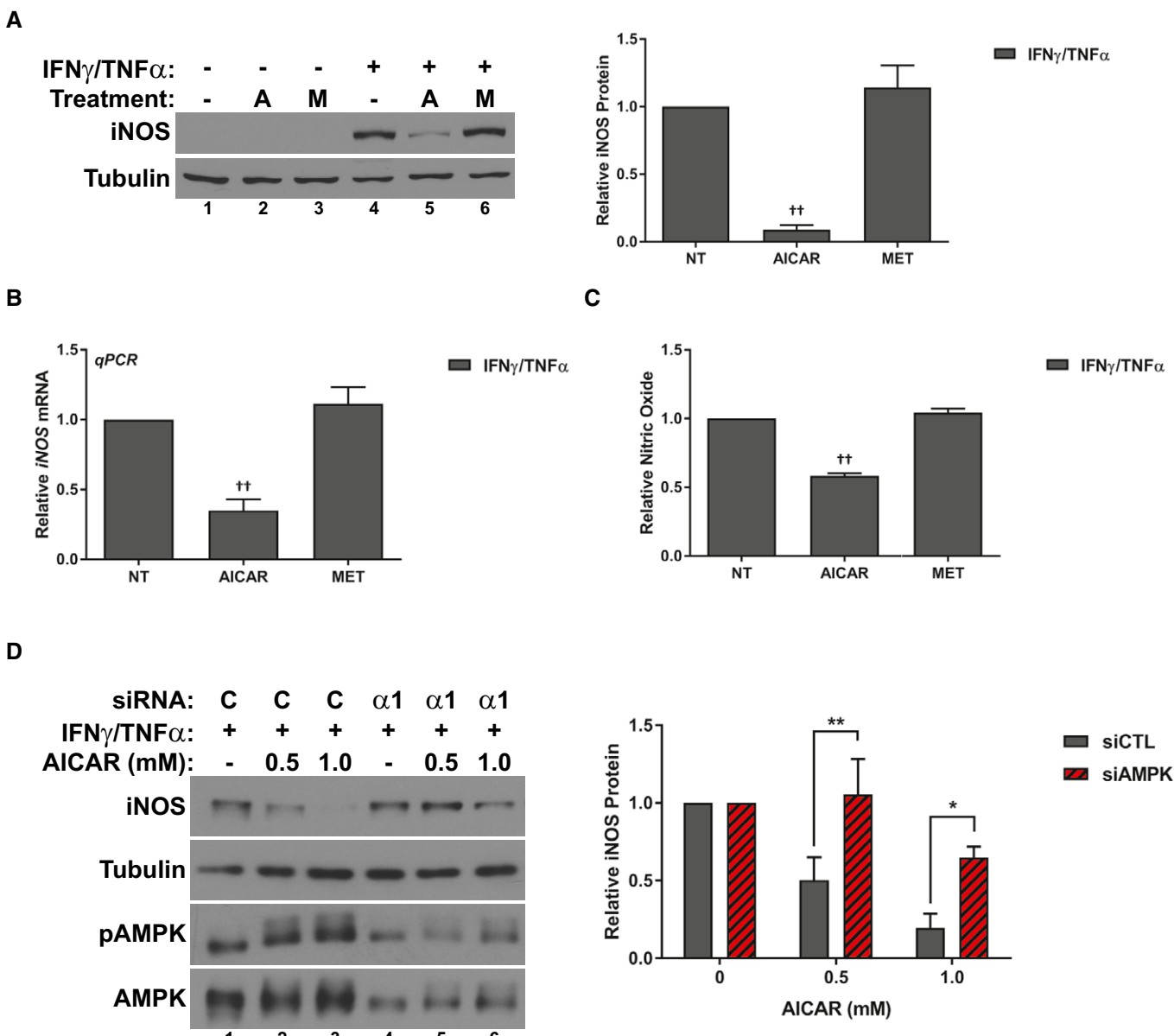

**Figure 4. The differential effects of AMPK agonists correlates with the expression of inducible nitric oxide synthase (iNOS).**

A   Western blotting for iNOS and tubulin protein levels 24 h after treatment. Quantification represents the levels of iNOS protein normalized to tubulin and relative to the IFNγ/TNFα control.

B   RT–qPCR analysis of iNOS mRNA levels relative to the IFNγ/TNFα control 24 h after treatment.

C   Nitric oxide (NO) levels in the culture media of cells relative to the IFNγ/TNFα control 24 h after treatment.

D   C2C12 myoblast cells were transfected with control (siCTL) or AMPKα1 (siAMPKα-1) targeting siRNA. Cells were subsequently treated with IFNγ/TNFα and the indicated doses of AICAR for 24 h. Western blotting for iNOS, tubulin, phospho-Thr172 AMPKα (pAMPK), and AMPK protein levels. Quantification represents the levels of iNOS protein normalized to tubulin and relative to the siCTL, IFNγ/TNFα control.

Data information: All quantifications are of three independent experiments (*n* = 3), and error bars represent the SEM. Significance between means was first determined using ANOVA. Significance *P*-values were calculated using Fisher's LSD. *$P < 0.05$; **$P < 0.01$ from equivalent siCTL samples; $^{††}P < 0.01$ from IFNγ/TNFα-treated controls. Source data are available online for this figure.

### AICAR treatment prevents the activation of the iNOS/NO pathway

In order to determine why these two different compounds, despite both activating AMPK, appear to have such varied effects, we began to look at how treatment with the different AMPK agonists affects the induction of inflammatory pathways during this process. We and others have previously reported that in response to inflammatory signaling, muscle significantly upregulates the expression of the iNOS enzyme, leading to NO release. Furthermore, treatments that

inhibit the iNOS/NO pathway are able to prevent cytokine-induced atrophy, suggesting that this pathway is a key downstream mediator of wasting (Buck & Chojkier, 1996; Di Marco et al, 2005, 2012; Ramamoorthy et al, 2009, Ma et al, 2017). Thus, we assessed the effect of treatment on the expression of iNOS and if this effect correlated with the ability of AICAR to prevent wasting. Indeed, AICAR, but not metformin, dramatically reduced iNOS protein and mRNA levels by 24 h (Fig 4A and B), with a corresponding decrease in the levels of NO in the culture media (Fig 4C). Interestingly, at higher doses, metformin does not affect iNOS expression and, in fact, induced myotube atrophy similarly to when myotubes are treated with inflammatory cytokines (Fig EV2). Therefore, these results suggest that one of the reasons for the differential effects of AICAR and metformin is their ability or inability to modulate iNOS expression, respectively.

## AICAR mechanism of action is likely dependent on AMPK

As described above, AICAR was found to activate AMPK much faster than cytokines (Fig EV1). Therefore, the beneficial effects of AMPK activation likely depend on the timing of activation. However, it is still unclear whether these effects of AICAR are in fact dependent on AMPK activity. To address this, we conducted knockdown experiments using siRNA targeting AMPK. Fully differentiated muscle cells express a large pool of AMPK. In addition, several isoforms of each subunit are present in muscle, making the AMPK system in muscle especially resistant to experimental attempts at knockdown (Steinberg & Kemp, 2009). Therefore, we conducted our studies in muscle pre-cursor cells (myoblasts), which express a lower level of AMPK and only one isoform of the catalytic subunit (AMPKα1). Importantly, while myoblasts cannot be assessed for atrophy, they maintain a similar response to inflammatory stimulus as seen in fully differentiated cells (Di Marco et al, 2005). Indeed, like myotubes, myoblasts will induce expression of iNOS and phosphorylation of AMPK 24 h after treatment with IFNγ/TNFα (Fig 4D). Therefore, we assessed whether the inhibition of iNOS by AICAR treatment could be impaired by AMPK knockdown in myoblasts. In myoblasts, knockdown of AMPKα1 prevented the ability of AICAR to inhibit iNOS, indicating that the effects on iNOS expression were AMPK-dependent (Fig 4D). In keeping with this, overexpression of human AMPKα1 increased sensitivity of myoblasts to AICAR treatment (Appendix Fig S1).

To further assess the AMPK dependency of the AICAR-mediated effects in differentiated myotubes, we conducted studies using the well-known AMPK inhibitor Compound C (Zhou et al, 2001). We first confirmed that Compound C treatment inhibited AICAR induced AMPK activation by measuring AMPK and ACC phosphorylation (Fig 5A). We then assessed the effect of AMPK inhibition on the anti-atrophic properties of AICAR. We found that co-treatment with Compound C negated the AICAR-mediated restoration of myotube widths in cytokine-treated myotubes (Fig 5B). This effect correlated with a 50% reduction in the recovery of normal glycolytic metabolism, as determined by glucose consumption and lactate production (Fig 5C and D). In keeping with this, co-treatment with Compound C completely impaired the ability of AICAR to restore anabolic signaling in cytokine-treated myotubes (Fig 5E). Finally, Compound C treatment was found to significantly reduce the inhibition of iNOS protein expression due to AICAR treatment (Fig 5F). Taken together, these results show that inhibition of AMPK signaling can blunt the effects of AICAR treatment on cytokine-induced myotube wasting.

If these AICAR-mediated effects are AMPK-dependent, other direct AMPK activators should be able to reproduce the effects of AICAR. To test this hypothesis, we conducted similar studies to those described above with another, more specific AMPK agonist, A-769662. A-769662 is a direct activator of AMPK that binds to the beta1 subunit to allosterically activate the kinase (Cool et al, 2006; Goransson et al, 2007; Sanders et al, 2007; Scott et al, 2008). Importantly, this mechanism, while still direct, is distinct from that of AICAR, as shown by the fact that co-treatment with both agonists has a synergistic effect (Ducommun et al, 2014). When we co-treated C2C12 myotubes with IFNγ/TNFα and A-769662, we found that A-769662 prevented myotube atrophy similarly to AICAR (Fig 6A). In addition, A-769662 replicated the effect of AICAR on aerobic glycolysis (Fig 6B and C), S6K/S6 phosphorylation (Fig 6D) and the activation of the iNOS/NO pathway (Fig 6E–G). Therefore, A-769662 treatment replicated the effects of AICAR on cytokine-induced atrophy.

As mentioned above, AICAR and A-769662 have been shown to act synergistically on AMPK by activating the kinase from distinct allosteric sites (Ducommun et al, 2014). Therefore, if the effects of these compounds are indeed through AMPK and not through different off-target pathways, they should be able to synergistically prevent the effects of cytokines on myotube wasting. To test this, we titrated co-treatments of sub-optimal doses of both AICAR and A-769662 and assessed their effect on cytokine-treated myotubes.

**Figure 5. Compound C inhibits the effects of AICAR on cytokine-induced atrophy.**

A    Western blotting for phospho-Thr172 AMPKα (pAMPK), total AMPKα (AMPK), phospho-Ser79-ACC (pACC), and total ACC 24 h after treatment. Quantification of the pAMPK/AMPK and pACC/ACC ratios relative to the non-treated (NT) control from four and three independent experiments, respectively (n = 4, 3).
B    Immunofluorescence staining for myoglobin and myosin heavy chain (MyHC) 48h after treatment. Scale bars represent 50 μm. Quantification represents the average myotube width.
C, D    Rates of glucose consumption (C) and lactate production (D) measured in the media 24 h after treatment relative to the NT control.
E    Western blotting of phospho-Thr389-p70S6K (pS6K), total p70S6K (S6K), phospho-Ser235/236-S6 (pS6), and total S6. Quantification represents the pS6K/S6K and pS6/S6 ratios relative to the NT control.
F    Western blotting for iNOS and tubulin protein levels 24 h after treatment. Quantification represents the levels of iNOS protein normalized to tubulin and relative to the IFNγ/TNFα control.

Data information: All quantifications are of three independent experiments (n = 3), unless otherwise stated, and error bars represent the SEM. Significance between means was first determined using ANOVA. Significance P-values were calculated using Fisher's LSD. *P < 0.05; **P < 0.01 from non-treated (NT) controls; †P < 0.05 from IFNγ/TNFα-treated controls.
Source data are available online for this figure.

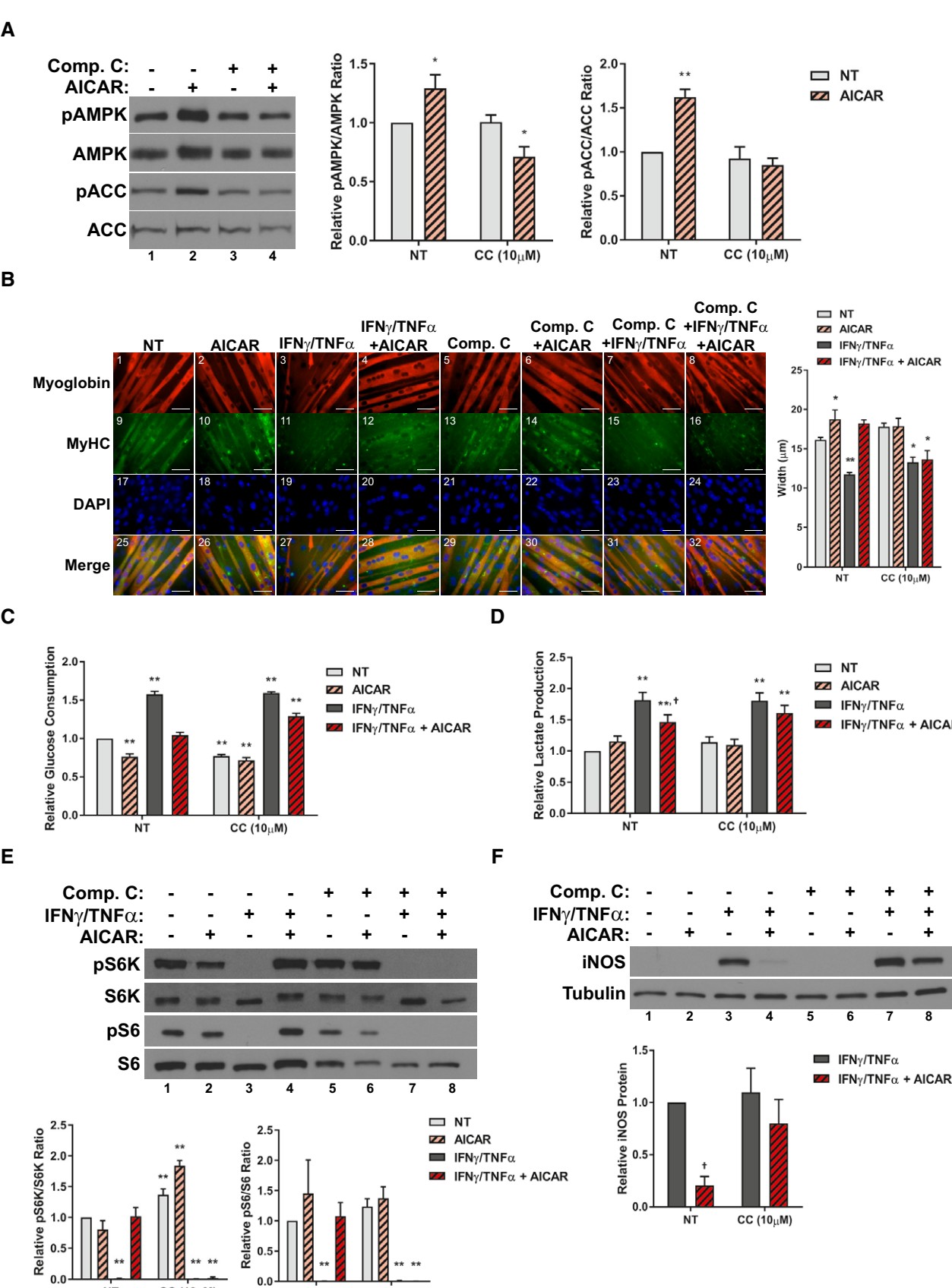

**Figure 5.**

Importantly, while treatment of these lower doses of the two compounds did not significantly affect myotube atrophy in response to IFNγ/TNFα alone, combined treatments significantly recovered myotube size (Fig 7A). Furthermore, co-treatment was able to synergistically inhibit the expression of iNOS (Fig 7B) and restore anabolic signaling (Fig 7C). Thus, AICAR and A-769662 can synergistically prevent cytokine-induced atrophy and signaling events, likely through AMPK. Taken together, the results of Compound C and A-769662 treatment and co-treatment strongly suggest that the effects of AICAR on inflammatory atrophy are mediated through direct AMPK activation.

## AICAR, but not metformin, prevents muscle atrophy in the C26 murine model of cancer cachexia

In order to assess whether the inhibition of cytokine-driven muscle wasting by AMPK is relevant in disease models of cachexia, we first performed *in vivo* studies in the BALB/C mouse strain. It has previously been shown that the subcutaneous injection of C26-adenocarcinoma cells into the flank of these mice leads to the development of cancer cachexia-like symptoms due to the production of pro-inflammatory cytokine, such as IL-6 (Tanaka *et al*, 1990). Both AICAR and metformin have been associated with anti-tumorigenic effects, and so the possibility exists that AICAR or metformin treatment could affect the growth of the C26 cells and, in doing so, prevent the onset and progression of cancer cachexia independent of muscle-specific effects (Jose *et al*, 2011; Kourelis & Siegel, 2012; Faubert *et al*, 2013). Indeed, daily intraperitoneal injections of AICAR or metformin at early stages of tumor development (day 9, when the C26 tumors were just palpable) triggered a robust reduction in tumor burden and subsequent muscle atrophy (Fig EV3). Therefore, we decided to begin treatment with these two agonists only after tumors had become well established (day 12) to assess the effects of these drugs on muscle loss independently from their effect on tumor growth. We observed that AICAR treatment, but not metformin, significantly reduced the extent of body weight loss in C26 bearing animals (Table 1). This recovery was associated with an improvement in overall musculature (Fig 8A) and an approximately 50% reduction in the extent of muscle mass loss, assessed in both the gastrocnemius and tibilalis anterior muscles (Table 1). However, there was no effect on adipose tissue wasting, assessed in the inguinal fat pad (Table 1). In this treatment regime, neither AICAR nor metformin significantly affected end-point tumor burden (Table 1). Serum levels of IL-6 were

elevated in all cancer-bearing mice, though there was a trend toward decreased levels in AICAR (non-significant) and metformin (significant)-treated mice (Table 1). There was no significant effect of AICAR or metformin treatment alone in non-tumor-bearing mice on body weight change or gastrocnemius weight (Appendix Fig S2A and B). Activation of AMPK in muscle was confirmed by detection of ACC phosphorylation, which was increased in mice treated with AICAR, metformin, or in cancer-bearing mice, as expected (Fig 8B). To confirm that AICAR treatment successfully reduced the extent of cancer-induced muscle atrophy, myofiber cross-sectional area (CSA) was determined in gastrocnemius muscle samples. C26 tumor-bearing animals had reduced CSA, which showed recovery with AICAR, but not metformin, following the same trend seen in the overall muscle weight (Fig 8C). In addition, the expression of the muscle-specific E3-ligase Atrogin-1/MAFbx was found to be decreased in the muscle of AICAR-, but not metformin-, treated mice and there was a non-significant trend toward decreased expression of MuRF1 (Fig 8D–E). Treatment with AICAR or metformin alone had no significant effect on the basal expression of Atrogin-1/MAFbx or MuRF1 (Appendix Fig S2C and D). Thus, AICAR, but not metformin, was found to reduce the extent of muscle wasting in the C26 model independent of tumor growth.

Given that the AICAR treatment was necessarily delayed to avoid affecting tumor growth, we hypothesized that AICAR treatment might be affecting the later stages of disease progression more strongly than the early stages. Indeed, we observed that the correlation between tumor growth and muscle weight loss in the gastrocnemius was lost in the AICAR-treated C26 cohort (Fig EV4A). Therefore, we assessed the extent of muscle loss in mice at day 14 and day 21 post-C26 inoculation. At day 14, mice have only received 2 days of AICAR treatment, in comparison with day 21, where they have received 9 days of treatment. Interestingly, we found that approximately half of the wasting observed at day 21 had already occurred by day 14, and that this loss was not affected after only 2 days of AICAR treatment (Fig EV4B). However, AICAR treatment significantly and dramatically reduced the further loss of muscle mass between day 14 and day 21 (Fig EV4B). Furthermore, assessment of Atrogin-1/MAFbx and MuRF1 expression at day 14 showed that AICAR treatment had significantly reduced the expression of Atrogin-1/MAFbx, but not MuRF1, in keeping with our previous observations (Fig EV4C). Therefore, it appears that there is significant portion of muscle loss that occurs prior to AICAR administration, but that AICAR effectively prevents the further progression of muscle wasting once it has been

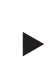

**Figure 6.  A-769662 prevents iNOS expression and is anti-cachectic.**

A    Immunofluorescence staining for myoglobin and myosin heavy chain (MyHC) 48 h after treatment. Scale bars represent 50 μm. Quantification represents the average myotube width.

B, C    Rates of glucose consumption (B) and lactate production (C) measured in the media 24 h after treatment relative to the DMSO-treated control.

D    Western blotting of phospho-Thr389-p70S6K (pS6K), total p70S6K (S6K), phospho-Ser235/236-S6 (pS6), and total S6. Quantification represents the pS6K/S6K and pS6/S6 ratios relative to the DMSO-treated control.

E    Western blotting of iNOS, tubulin, phospho-Thr172 AMPKα (pAMPK), total AMPKα (AMPK), phospho-Ser79-ACC (pACC), and total ACC. Quantification represents the levels of iNOS protein normalized to tubulin and relative to the IFNγ/TNFα control.

F    RT–qPCR analysis of iNOS mRNA levels relative to the IFNγ/TNFα control 24 h after treatment.

G    Nitric oxide (NO) levels in the culture media of cells relative to the IFNγ/TNFα control 24 h after treatment.

Data information: All quantifications are of three independent experiments ($n = 3$), and error bars represent the SEM. For panels (A–D), significance between means was first determined using ANOVA. Significance *P*-values were calculated using Fisher's LSD. For panels (E–G), *P*-values were calculated using the Student's *t*-test. *$P < 0.05$; **$P < 0.01$ from DMSO controls; ††$P < 0.01$ from IFNγ/TNFα-treated controls.

Source data are available online for this figure.

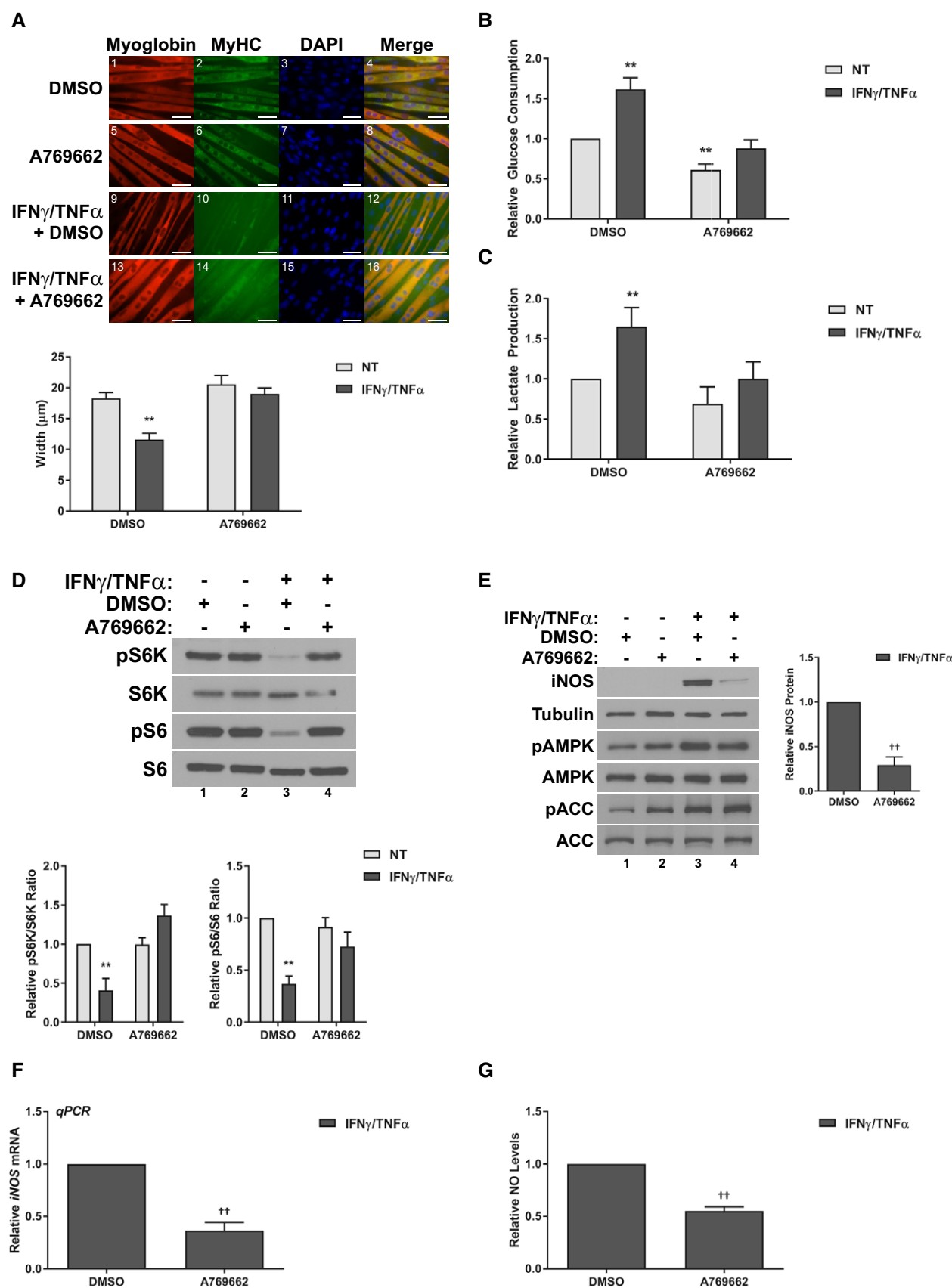

**Figure 6.**

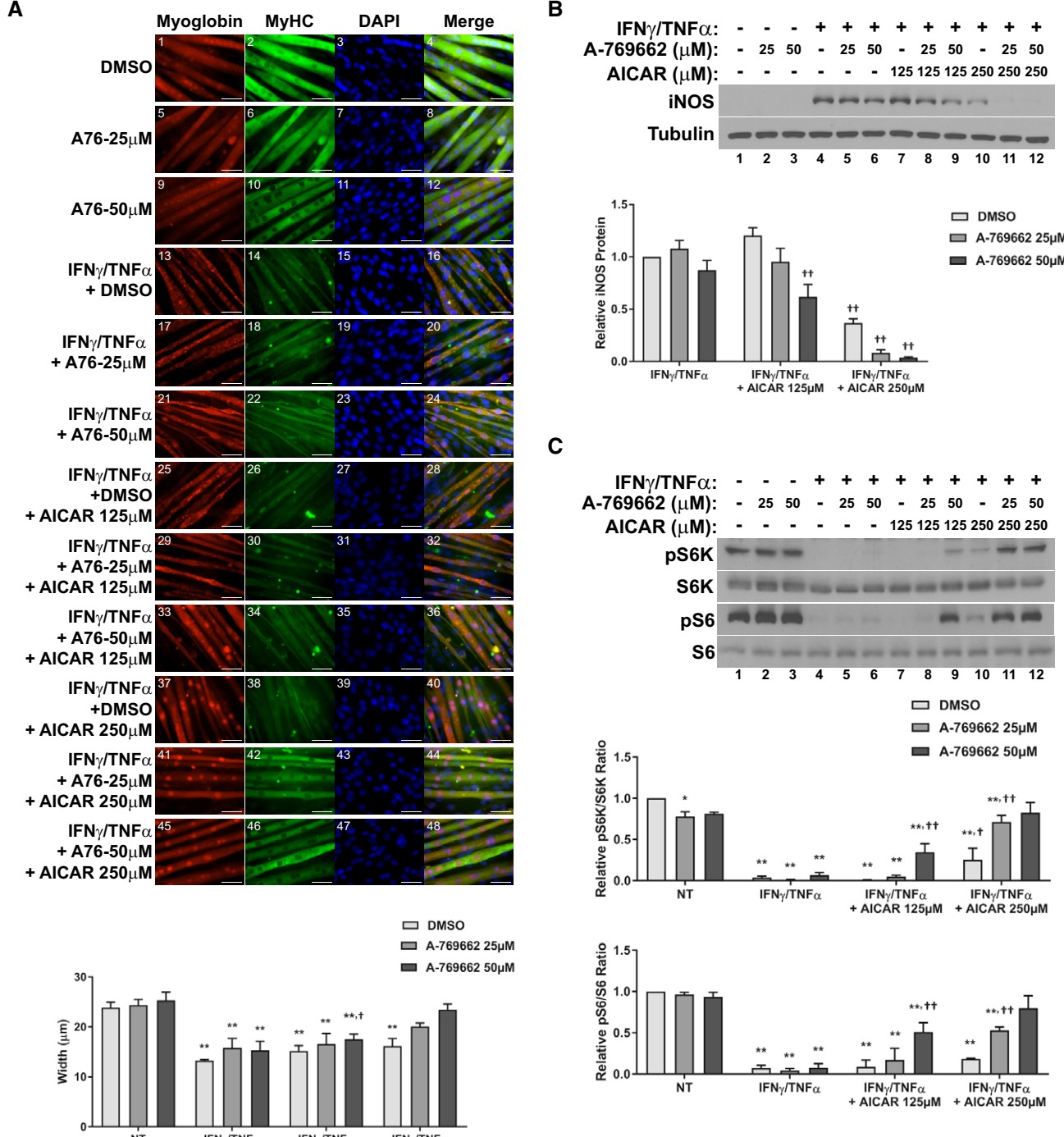

**Figure 7. AICAR and A-769662 synergistically prevent cytokine-induced myotube atrophy.**

A   Immunofluorescence staining for myoglobin and myosin heavy chain (MyHC) 48 h after treatment. Scale bars represent 50 μm. Quantification represents the average myotube width.

B   Western blotting for iNOS and tubulin protein levels 24 h after treatment. Quantification represents the levels of iNOS protein normalized to tubulin and relative to the IFNγ/TNFα control.

C   Western blotting of phospho-Thr389-p70S6K (pS6K), total p70S6K (S6K), phospho-Ser235/236-S6 (pS6), and total S6. Quantification represents the pS6K/S6K and pS6/S6 ratios relative to the NT control.

Data information: All quantifications are of three independent experiments ($n$ = 3), and error bars represent the SEM. Significance between means was first determined using ANOVA. Significance *P*-values were calculated using Fisher's LSD. *$P < 0.05$; **$P < 0.01$ from non-treated (NT) controls; †$P < 0.05$; ††$P < 0.01$ from IFNγ/TNFα-treated controls.

Source data are available online for this figure.

**Table 1. AICAR, but not metformin, mitigates muscle atrophy in the C26 murine model of cancer cachexia.**

|  | Saline | C26 | C26 + AICAR | C26 + Metformin |
|---|---|---|---|---|
| Body Weight Change (%) | $5.0 \pm 0.4$ | $-19.0 \pm 1.7^{**}$ | $-9.6 \pm 2.5^{**,\dagger\dagger}$ | $-16.3 \pm 1.5^{**}$ |
| Tibialis Anterior (mg) | $37.6 \pm 2.2$ | $26 \pm 1.5^{**}$ | $31.3 \pm 1.4^{*,\dagger}$ | $25.4 \pm 0.7^{**}$ |
| Gastrocnemius (mg) | $122.6 \pm 2.1$ | $87.3 \pm 3.9^{**}$ | $101.8 \pm 2.7^{**,\dagger}$ | $83.8 \pm 4.6^{**}$ |
| Fat Pad (mg) | $137 \pm 6.9$ | $10.9 \pm 6.4^{**}$ | $15 \pm 6.6^{**}$ | $11.8 \pm 3.1^{**}$ |
| Tumor (g) | NA | $1.024 \pm 0.060$ | $1.038 \pm 0.104$ | $1.007 \pm 0.065$ |
| $n$ | 9 | 10 | 6 | 8 |
| IL-6 (pg/ml) ($n = 4$) | $30.8 \pm 1.7$ | $253.9 \pm 46.1^{**}$ | $201.9 \pm 55.2^{*}$ | $127.7 \pm 34.9^{\dagger}$ |

Values represent the mean $\pm$ SEM.
$*P < 0.05$, $**P < 0.01$ compared to Saline cohort.
$^{\dagger}P < 0.05$, $^{\dagger\dagger}P < 0.01$ compared to C26 cohort.

administered to C26 tumor-bearing mice. It is likely, then, that earlier treatment with AICAR could be more beneficial, but due to the limitation of having to treat later to prevent the effects of AICAR on tumor growth, this was not possible in our model. Of note, this limitation would not be present in the clinical context, as both the anti-tumorigenic and anti-cachectic properties of AMPK activation early on in disease progression would be beneficial for the treatment of cancer cachexia. Nevertheless, results from this model show that AICAR administration can effectively prevent muscle wasting in the C26 cancer cachexia model, independent of effects on tumor progression.

### AICAR, but not metformin, prevents muscle atrophy induced by IFNγ/TNFα injection

It is possible that the sparing of muscle mass in AICAR treatment is due to systemic effects, that impact muscle wasting indirectly (Galic *et al*, 2011; Mounier *et al*, 2013; O'Neill & Hardie, 2013). Indeed, both AICAR and metformin treatment showed decreased circulating IL-6, which is believed to be a driver of the cachectic phenotype in the C26 model (Aulino *et al*, 2010). However, given that metformin treatment reduced IL-6 levels more significantly than AICAR treatment but had no effect on muscle mass, it is likely that the levels of IL-6 seen in AICAR and metformin treatment are sufficient to induce cachexia. To further confirm that the effects of AICAR are specific to the interaction between inflammation and muscle, and not due to indirect effects on the tumor or circulating cytokine levels, we conducted wasting evaluations in mice that were intramuscularly injected with the inflammatory cytokines IFNγ/TNFα. We have previously shown that this tumor-free model can induce muscle wasting (Di Marco *et al*, 2005, 2012; Ma *et al*, 2017). Importantly, because the cytokines are administered directly to the muscle tissue, it is not possible for AICAR treatment to affect the initial inflammatory stimulus in this system. In this model, AICAR treatment had no effect on total body weight changes, but prevented muscle mass loss (Fig EV5). Therefore, it is likely that AICAR treatment can prevent muscle wasting, at least in part, by affecting the response to inflammation at the site of muscle tissue.

### AICAR, but not metformin, prevents muscle atrophy in a murine model of septic cachexia

To further assess the efficacy of AICAR in models of inflammation-associated muscle wasting, we next assessed a model of septic

cachexia. Intraperitoneal injection of LPS has been shown to induce a septic-like state in mice. While the immune response to LPS in mice does not completely reflect that seen in humans, it does induce the expression of TNFα and IL-6 in mice, and has previously been shown to induce muscle loss and the expression of the atrophic muscle E3 ligases, similar to human sepsis-associated cachexia, within 24 h (Copeland *et al*, 2005; Jin & Li, 2007; Callahan & Supinski, 2009; Doyle *et al*, 2011; Braun *et al*, 2013). Therefore, we co-injected mice with LPS and either AICAR or metformin and assessed the effect on muscle wasting. It has previously been reported that LPS-injected mice have significantly reduced food consumption rates (Braun *et al*, 2013). Therefore, we pair-fed our controls (PF) and our LPS and AICAR or metformin co-treated cohorts to the LPS cohort consumption rate in order to account for any variation that might have arisen from differences in food consumption. We observed that LPS mice have a significant reduction in body weight following LPS injection (Table 2). However, this body weight loss was largely accounted for by the decrease in food consumption, as observed by the fact that the PF cohort showed similar body weight loss (Table 2). Unlike in the C26 model, this acute model of inflammation did not significantly affect the weight of adipose tissue (Table 2). However, there was a significant loss of muscle mass in both the gastrocnemius and tibialis anterior in LPS-treated mice when compared to PF controls, which was completely prevented by AICAR, but not metformin, treatment (Table 2). The effects on muscle atrophy were further confirmed by cross-sectional area analysis of the gastrocnemius muscle. As for the muscle weights, we observed that LPS-injected mice had significantly reduced average muscle fiber CSA, which was completely abrogated by AICAR treatment and unaffected by metformin treatment (Fig 9A). Furthermore, AICAR treatment, but not metformin, was able to prevent the increased expression of both Atrogin-1/MAFbx and MuRF1 in the skeletal muscle of LPS-injected mice (Fig 9B and C). Treatment with AICAR or metformin alone did not significantly affect body weight change, muscle mass, or E3 ligase expression, though there was a non-statistically significant trend toward decreased MuRF1 expression in the metformin control cohort (Appendix Fig S3). However, in the LPS and metformin co-treated cohort, this reduction was not observed (Fig 9C). Thus, as in our other models, only AICAR prevented muscle wasting in a model of septic cachexia.

Taken together, the results from these animal studies show that AICAR can mitigate or prevent muscle atrophy in a variety of inflammatory disease contexts, including both chronic (cancer) and

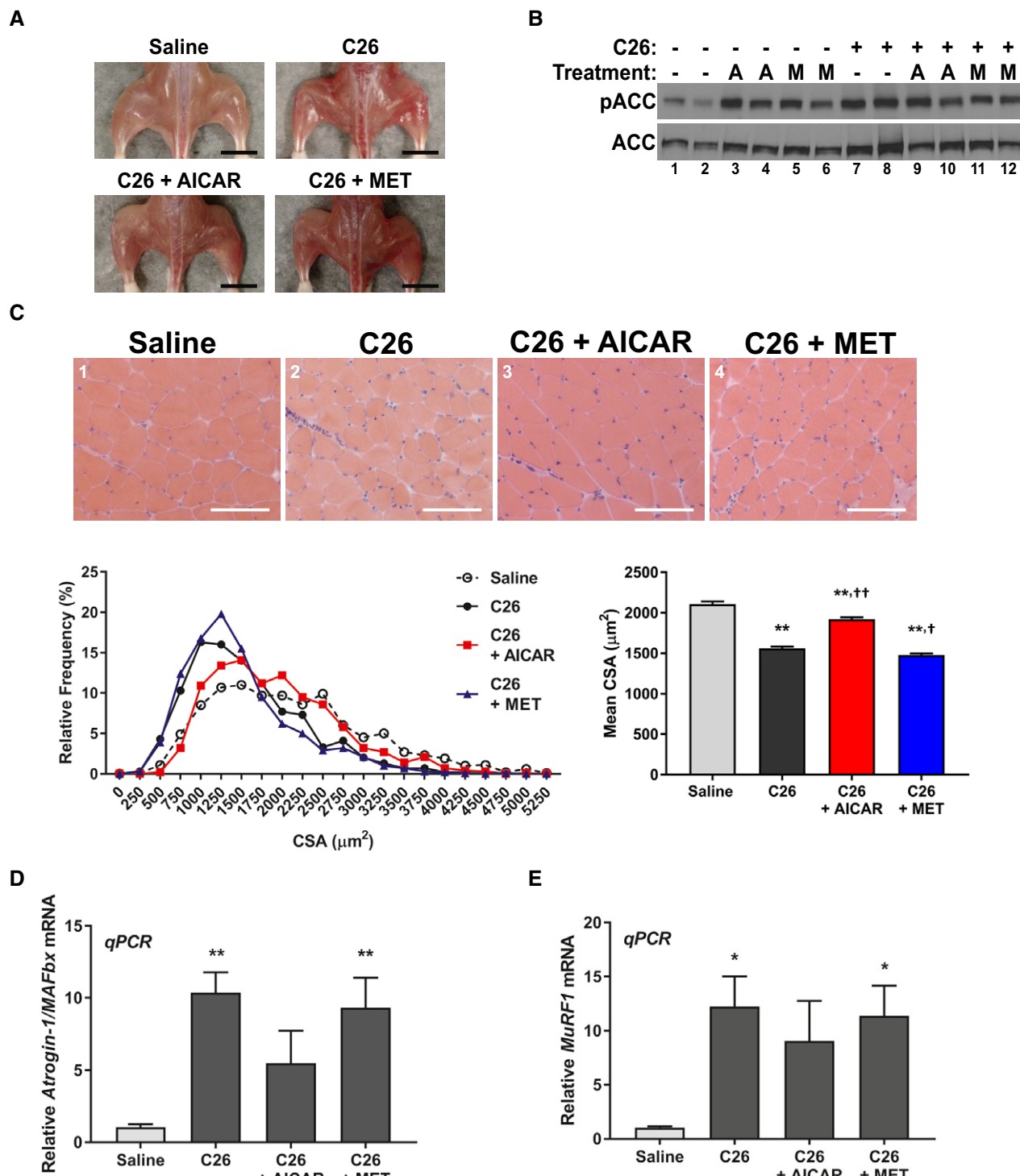

**Figure 8. AICAR partially recovers muscle mass in the C26-mouse model of cancer cachexia.**

A    Representative images of overall hind limb musculature. Scale bar represents 1 cm.

B    Western blotting for phospho-Ser79-ACC (pACC) and total ACC.

C    Cross-sectional analysis of the gastrocnemius muscle calculated from 250 fibers per mouse from four mice per cohort (n = 4). Cross-sections were stained by H&E. Scale bars represent 100 µm.

D, E    RT–qPCR analysis of Atrogin-1/MAFbx (D) and MuRF1 (E) mRNA expression from the tibialis anterior of three mice per cohort (n = 3).

Data information: Error bars represent the SEM. Significance between means was first determined using ANOVA. Significance P-values were calculated using Fisher's LSD.
*P < 0.05; **P < 0.01 from saline controls; †P < 0.05; ††P < 0.01 from C26 controls.
Source data are available online for this figure.

Table 2. AICAR, but not metformin, mitigates muscle atrophy in the LPS model of septic cachexia.

| | Vehicle | PF | LPS | LPS + AICAR | LPS + Metformin |
|---|---|---|---|---|---|
| Body Weight Change (%) | 2.1 ± 0.2** | −9.5 ± 1.3 | −10.4 ± 0.7 | −6.5 ± 1* | −11.4 ± 0.4 |
| Tibialis Anterior (mg) | 44.1 ± 3.4 | 43.2 ± 0.8 | 38.3 ± 1.5** | 42.2 ± 0.9 | 39.2 ± 1.2* |
| Gastrocnemius (mg) | 156.5 ± 4.8 | 145.7 ± 3.2 | 136.3 ± 3.6* | 144.8 ± 2.9 | 136.5 ± 2.2* |
| Fat Pad (mg) | 99.7 ± 10.6 | 97.6 ± 10.3 | 106.1 ± 16.8 | 95.5 ± 8.1 | 107.7 ± 12.1 |
| *n* | 4 | 11 | 11 | 11 | 11 |

Values represent the mean ± SEM.
*$P < 0.05$, **$P < 0.01$ compared to the pair-fed (PF) cohort.

acute (sepsis, cytokine injection) conditions. This suggests that AMPK agonists that do not affect mitochondrial function could be used as novel therapies, in conjunction with other treatments, to prevent the loss of muscle mass in patients with chronic or acute inflammatory conditions.

## Discussion

While AMPK has been associated with the progression of cachexia, its anti-inflammatory properties suggest that AMPK activation could be anti-cachectic, since inflammation is the primary driver of cachectic muscle wasting (Tisdale, 2009). To understand whether AMPK activation could prevent inflammation-driven atrophy, we tested the effects of drug-induced AMPK activation in models of inflammatory muscle wasting. Interestingly, we found that the AMP-mimic AICAR, but not mitochondrial inhibitor metformin, prevented inflammation-associated atrophy in both *in vitro* and *in vivo* models, despite both being commonly used as AMPK activators (Figs 1, 8, 9, and EV4; Grahame Hardie, 2016). Prevention of atrophy correlated with restoration of oxidative metabolism and anabolic signaling, as well as impairment of one of the known effector of muscle wasting, the iNOS/NO pathway (Figs 2–4). Inhibition of AMPK activation with Compound C was found to block the protective effects of AICAR (Fig 5). In addition, a more specific AMPK activator, A-769662, which activates AMPK through the beta subunit, was found to replicate the effects of AICAR (Fig 6). AICAR and A-769662 were also found to be able to act synergistically to prevent atrophy in myotubes (Fig 7). These results suggest that the beneficial effects of AICAR in cytokine-driven atrophy are likely to be AMPK-dependent. However, we also found that AMPK was activated by cytokines alone and metformin treatment. In these conditions, AMPK activation was instead associated with mitochondrial dysfunction, as well as impairment of mTOR and anabolism, in keeping with previous reports (White *et al*, 2011, 2013; Figs 3 and EV2). Together, these observations indicate that the role of AMPK in inflammatory-driven muscle atrophy is complex and likely context dependent.

The different outcomes of AMPK activation observed in our study are likely linked to the nature and timing of the activation. Our results indicate that cytokine-associated muscle wasting is accompanied by an elevation in the glycolytic rate and severely impaired mitochondrial oxidative respiration (Fig 2). Importantly, metformin treatment alone had many of the same effects as inflammatory cytokine treatment. This is not surprising, as metformin has been shown to directly inhibit complex I in the

electron transport chain (Andrzejewski *et al*, 2014; Bridges *et al*, 2014). Indeed, the activation of AMPK by metformin is believed to be an indirect consequence of the metabolic stress induced by inhibition of oxidative phosphorylation and the resulting increase in cytoplasmic AMP levels (Grahame Hardie, 2016). Likely, this is also the mechanism behind the activation of AMPK during cytokine treatment alone. Importantly, we observed that both low and high doses of metformin induced wasting similar to cytokine treatment in C2C12 myotubes (Figs 1C and EV2). We also observed a reduction in both muscle mass (non-significant, Table 1) and CSA (significant, Fig 8C) in the C26 and metformin co-treated cohort when compared to the C26-treated cohort. Therefore, it would appear that the mitochondrial toxicity of metformin supersedes any potential beneficial effects of the activation of AMPK in the context of inflammation-associated atrophy, and it is possible that metformin treatment during inflammatory disease may even be detrimental to muscle mass, to an extent. From this, we conclude that AMPK activation during or after the induction of mitochondrial dysfunction, as is the case in cytokine and metformin treatment, but not AICAR, does not prevent muscle atrophy and likely contributes to the suppression of anabolic signaling through mTOR, as has previously been suggested (White *et al*, 2013). This activation, then, is likely to represent a later stage of inflammatory atrophic signaling. Consistent with this, we observed that AMPK is activated in the later stages of inflammatory signaling in muscle, both *in vitro* and *in vivo* (Figs 1A, 8B, and EV1). This observation is corroborated by a previous study in the Apc[Min/+] colon cancer model that showed that AMPK activation occurs in the later stages of cachectic progression (White *et al*, 2011). In contrast, ZMP, the active form of AICAR, and A-769662 both activate AMPK directly by binding to the gamma and beta subunits, respectively, and are not dependent on the energetic and metabolic status of the cell (Grahame Hardie, 2016). As a result, we observed that AICAR treatment was able to induce AMPK activation much earlier than cytokine treatment (Fig EV1). This early activation of AMPK by AICAR and/or A-769662, before the induction of metabolic stress, was associated with both a protection of metabolic function (Fig 2) and the prevention of both myotube and muscle wasting (Figs 1 and 6–9). Therefore, we propose that the activation of AMPK before the induction of metabolic stress, as is the case with AICAR and A-769662, can protect muscle from inflammation-induced atrophy, but that, induction of AMPK activity by metabolic stress, as is the case with metformin and cytokines, instead contributes to the suppression of anabolism.

One apparent contradiction that arises from this duality of AMPK function in inflammatory muscle wasting is the apparent ability for AMPK activation to be associated with both the suppression and

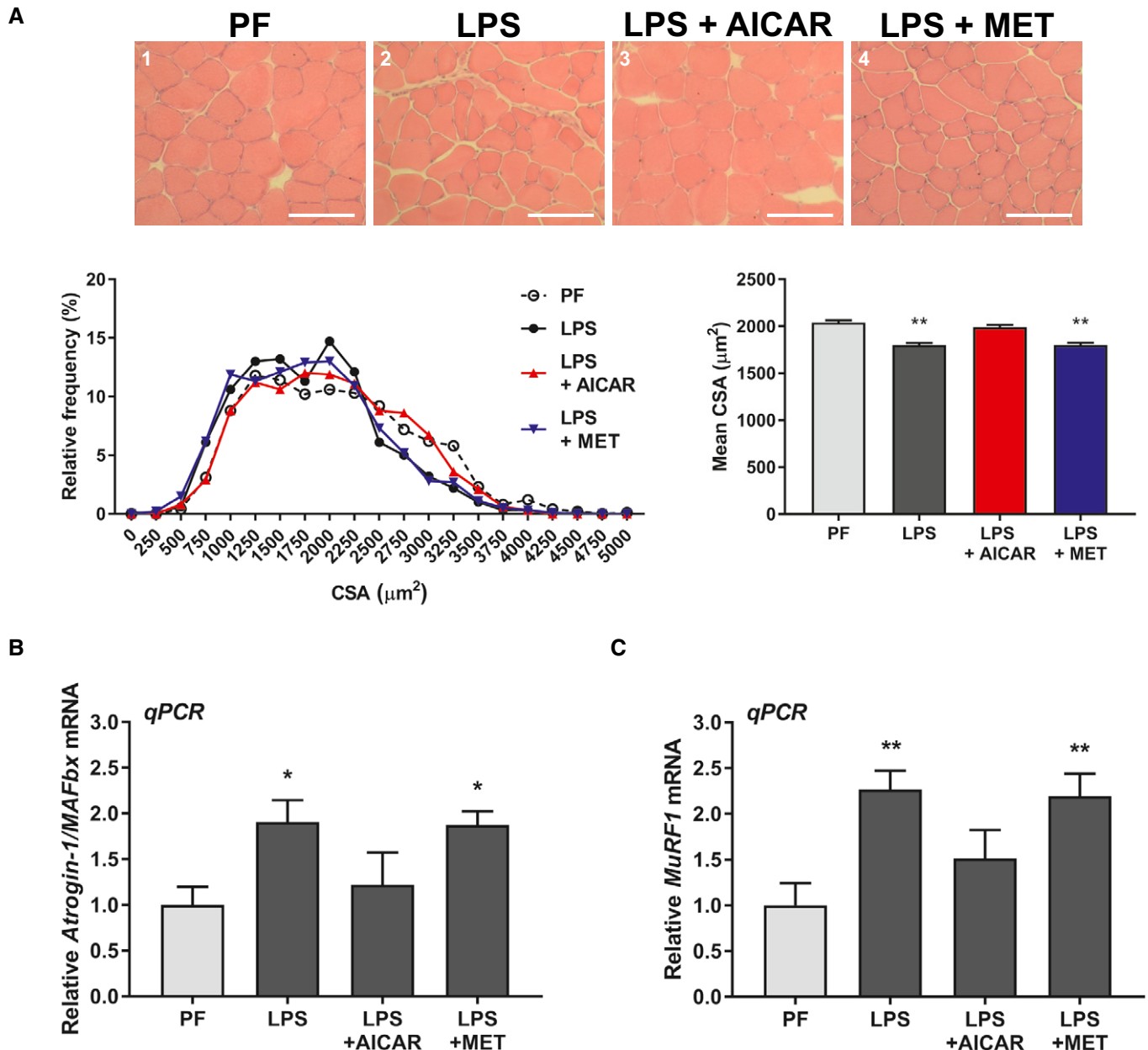

**Figure 9. AICAR completely prevents muscle wasting in response to LPS injection.**

A   Cross-sectional analysis of the gastrocnemius muscle calculated from 250 fibers per mouse from four mice per cohort (*n* = 4). Cross-sections were stained by H&E. Scale bars represent 100 μm.

B, C   RT–qPCR analysis of Atrogin-1/MAFbx (B) and MuRF1 (C) mRNA expression from the tibialis anterior of four mice per cohort (*n* = 4).

Data information: Error bars represent the SEM. Significance between means was first determined using ANOVA. Significance *P*-values were calculated using Fisher's LSD. *$P < 0.05$; **$P < 0.01$ from PF controls.

Source data are available online for this figure.

recovery of anabolic signaling through mTOR and protein synthesis. Indeed, we observed that activation of AMPK during cytokine and metformin treatment was associated with a hypo-phosphorylation of both S6 kinase and S6 and a decreased rate of protein synthesis (Figs 3 and EV2). As described above, this suppression has been linked to AMPK activity (White *et al*, 2011, 2013). However, treatment with AICAR and/or A-769662 recovered S6K and S6

phosphorylation and protein translation, despite the fact that AMPK activity is known to inhibit mTOR (Inoki *et al*, 2012). Importantly, the doses of AICAR and A-769662 used in our studies were not associated with significant or persistent suppression of anabolic signaling when used alone (Figs 3 and 6). Thus, we do not believe that AICAR or A-769662-induced AMPK activation directly affects mTOR signaling in our model, but rather indirectly restores it by inhibiting

the upstream inflammatory events that eventually lead to anabolic repression. This suggests, again, that the context of AMPK activation during cachexia is likely to be critical, and that a therapeutic window exists for the use of AMPK agonists in cachexia. It is likely that activating AMPK would only be effective at the onset or early stages of cachexia development, before the induction of metabolic stress or anabolic signaling suppression. In contrast, in later stages of cachexia, AMPK inhibition might prove to be more therapeutically beneficial.

In addition to effects on metabolism and anabolic signaling, we found that the ability of the different AMPK activators to prevent, or not, the progression of muscle wasting correlated with inhibition of the expression of the pro-cachectic gene iNOS. We and others have previously shown that this enzyme plays a very important role as a downstream effector of inflammation-driven wasting (Buck & Chojkier, 1996; Di Marco *et al*, 2005, 2012; Ramamoorthy *et al*, 2009; Hall *et al*, 2011, Ma *et al*, 2017). Also, we have previously shown that drugs that inhibit this enzyme can prevent cachexia, further highlighting the therapeutic relevancy of inhibiting iNOS expression (Di Marco *et al*, 2005, 2012; Ma *et al*, 2017). Therefore, it is likely that inhibition of iNOS is a key mechanism by which AMPK activation can prevent inflammation-driven muscle wasting.

A possible alternative explanation for the observations in this study is that the effects of AICAR arise from an off-target pathway and are AMPK independent. Indeed, AICAR treatment is known to activate AMPK-independent pathways (Quan *et al*, 2015; Rao *et al*, 2016). However, we found that the effects of AICAR treatment on atrophy, metabolism, mTOR signaling, and iNOS expression were all prevented by treatment with the AMPK inhibitor Compound C (Fig 5). Importantly, Compound C has been described to inhibit inflammatory signaling independent of AMPK inhibition (Kim *et al*, 2011; Rao *et al*, 2016). If this effect was dominant in our model, Compound C treatment would be expected to enhance AICAR-mediated inhibition of the inflammatory phenotype. However, we observed the opposite effect and found that Compound C impaired the anti-inflammatory effects of AICAR. Given this, it is likely that the observed effects of Compound C are dependent on the inhibition of AMPK activity. To further confirm that the effects of AICAR were AMPK dependent, we also tested A-769662, a more specific AMPK activating compound (Cool *et al*, 2006; Goransson *et al*, 2007; Sanders *et al*, 2007; Scott *et al*, 2008). We found that A-769662 had the same effects as AICAR in our models (Fig 6). Importantly, A-769962 and the AICAR derivative ZMP have distinct chemical structures and bind to different subunits of AMPK heterotrimers (Grahame Hardie, 2016). In fact, A-769662 and AICAR have been shown to be able to activate AMPK synergistically (Ducommun *et al*, 2014). Consistent with this, our data show that co-treatments of sub-optimal doses of both AICAR and A-769662 were able to reverse the effects of cytokine treatment on myotubes (Fig 7). Thus, given their chemical dissimilarity and the fact that co-treatment of the two compounds synergistically inhibits cytokine-induced myotube atrophy, it seems unlikely that the effects observed in our study can be explained by the convergence of separate off-target effects of these compounds. Therefore, there is strong evidence to suggest that the effects of AICAR and A-769662 treatment on inflammation-induced muscle wasting are through their ability to activate AMPK.

We found that AICAR administration was effective at preventing muscle atrophy in a variety of mouse models of cachexia. In the C26

model of cancer cachexia (Table 1, Fig 8) and the LPS model of septic cachexia (Table 2, Fig 9), as well as a model of cytokine-induced atrophy (Fig EV5), we found that AICAR, but not metformin prevented muscle loss. However, we did not see any effects on adipose tissue wasting, another important characteristic of cachectic wasting, despite a previous study that found that an AMPK stabilizing peptide was effective at preventing adipose tissue wasting in cachexia (Table 1; Rohm *et al*, 2016). This discrepancy is most likely due to the fact that AMPK protein levels are significantly depleted in cachectic adipose tissue, but not in skeletal muscle (Fig 8; Rohm *et al*, 2016). Therefore, administration of an AMPK agonist alone would be insufficient to recover adipose mass, as there would be no molecular target in said tissue. This suggests that the dual administration of an AMPK stabilizer and an AMPK agonist may effectively recover both lean muscle and adipose tissue mass during cachectic wasting.

Cachexia is a multifactorial disease, requiring a multimodal therapeutic approach. Our study strongly suggests that AMPK activators could represent a novel therapeutic development space for the treatment of a variety of cachectic conditions. Indeed, the therapeutic potential of AMPK activation is further unscored by the fact that AMPK activation can protect muscle in *mdx* mice (a model of Duchenne's Muscular Dystrophy) and in Angiotensisin II-driven muscle atrophy (Tabony *et al*, 2011; Pauly *et al*, 2012). However, our findings, as discussed above, also indicate that there is likely a therapeutic window early on in the cachectic molecular pathophysiology during which AMPK activation is effective. In contrast, in the later stages of disease, AMPK inhibition might prove to be more beneficial, as has previously been suggested (White *et al*, 2013). It is clear that the role of AMPK in muscle-related disease is complex, and more studies are needed to further understand how and when AMPK can be beneficial or detrimental, and when it is activation or inhibition could be used therapeutically. Greater understanding in this area could then lead to the development of novel therapeutics that could be used in concert with other treatments to combat the deadly cachexia syndrome.

# Materials and Methods

### Reagents and antibodies

IFN$\gamma$ and TNF$\alpha$ were purchased from R&D Systems. AICAR and metformin were obtained from Toronto Research Chemicals (TRC) and Sigma-Aldrich, respectively. A-769662 and Compound C were obtained from TOCRIS Bioscience. Antibodies against pThr172-AMPK$\alpha$ (#2535), AMPK$\alpha$ (#2603), pSer79-ACC (#3661), ACC (#3662), pThr389-pS6K (#9205), S6K (#2708), pSer235/236-S6 (#2211), and S6 (#2317) were obtained from Cell Signaling Technology. Anti-iNOS antibody (#610431) was obtained from BD Transduction Laboratories. The anti-tubulin antibody (DSHB Hybridoma Product 6G7; deposited by Halfter, W.M.) and the anti-myosin heavy-chain antibody (DSHB Hybridoma Product MF20; deposited by Fischman, D.A.) were obtained from the Developmental Studies Hybridoma Bank, created by the NICHD of the NIH and maintained at The University of Iowa, Department of Biology, Iowa City, IA 52242. Anti-myoglobin antibody (ab77232) was obtained from Abcam.

## Cell culture and treatment

C2C12 myoblast cells (ATCC, Manassas, VA, USA) were grown and maintained in Dulbecco's modified Eagle's medium (DMEM; Invitrogen) containing 20% fetal bovine serum (Sigma) and 1% penicillin/streptomycin (P/S) antibiotics (Invitrogen). Cells were routinely monitored for mycoplasma infection by DAPI staining. To differentiate, C2C12 cells were allowed to reach 90–100% confluence on plastic coated with a 0.1% gelatin solution and were then switched to DMEM containing 2% horse serum (Invitrogen) and 1% P/S. On the third or fourth day following induction of differentiation, myotubes were treated or not with IFNγ (100 U/ml) and TNFα (20 ng/ml; Di Marco et al, 2012, 2005; Ma et al, 2017). At the same time, cells were treated or not with either AICAR (0.5 mM), metformin (MET; 0.5 mM), or A-769662 (100 μM; Goransson et al, 2007; Zhou et al, 2001).

## Animal models

All animal experiments were carried out with approval from the McGill University Faculty of Medicine Animal Care Committee and are in accordance with the guidelines set by the Canadian Council of Animal Care. Animals were randomly assigned to each treatment group by cage at the time of the experiment.

## Cancer cachexia model

Male BALB/C mice, age 4–6 weeks weighing on average 23 g, were obtained from the Jackson Laboratory. Mice were injected subcutaneously in the right flank with either C26 colon cancer cells at $1.0–1.5 \times 10^6$ cells per animal or an equal volume of saline solution. Tumor growth was monitored by measuring tumor surface area every other day with calipers. On day 9 or day 12 post-injection, animals were injected intraperitoneally with either AICAR (500 mg/kg/day), metformin (250 mg/kg/day), or an equal volume of saline. Mice were injected every day thereafter at approximately the same time for the next 7 days. This dose of AICAR and metformin equate to a human equivalent dose of approximately 40 mg/kg/day and 20 mg/kg/day when accounting for surface area and are comparable to the doses reported in clinical trials using these compounds for cancer treatment (Bost et al, 2016; Nair & Jacob, 2016). On day 19–21 post-C26 cell injection, animals were euthanized by $CO_2$ asphyxiation followed by cervical dislocation. Immediately following death, mice were exsanguinated and serum was frozen at −80°C. The muscle tissue and fat pad of each animal were then dissected and weighed immediately. After, muscles to be used for histological analysis were frozen in liquid nitrogen cooled isopentane (Sigma) and stored at −80°C. All other tissues were snap-frozen directly in liquid nitrogen.

## LPS septic cachexia model

The LPS model of septic shock was performed on male C57Bl/6 mice age 8–12 weeks, weighing on average 26 g, obtained from Jackson Laboratory. LPS was prepared fresh the day of the experiment in a solution of 0.5% bovine serum albumin (BSA) in phosphate-buffered saline (PBS). LPS solution was then diluted with either additional saline for control cohorts or a PBS solution of either AICAR or metformin. The final dose of LPS administered was 1 mg/kg, which as previously been reported to induce muscle atrophy and muscle-specific E3 ligase expression (Jin & Li, 2007; Doyle et al, 2011; Braun et al, 2013). The final dose of AICAR was 500 mg/kg, and the final dose metformin was 250 mg/kg. Mice were injected intraperitoneally with the appropriate combination of compounds at the beginning of the dark cycle, between 18:30 and 19:30. All mice were separated into individual cages (1 mouse per cage) to allow for pair-feeding and food consumption monitoring. For pair-fed cohorts, mice were given the average food consumed by a pilot cohort of LPS-injected mice. Mice were euthanized the following day, approximately 18h after initial injection, as described above.

## Intramuscular cytokine injection mouse model of atrophy

For intramuscular cytokine injection experiments, male C57Bl/6 mice age 4–6 weeks, weighing approximately 23 g were obtained from the Jackson Laboratory. Every day for 5 days, mice were intraperitoneally injected with either AICAR (350 mg/kg) or an equivalent volume of saline. One hour after, mice were intramuscularly injected with a cocktail of IFNγ/TNFα (7,500 U/mouse; 3 μg/mouse) or an equivalent volume of saline in the upper hind limb muscles. On the final day of injection, 3–4 h after the final injection of cytokines, mice were euthanized and tissues were weighed and collected as described in the C26 model of cancer cachexia.

## Histological analysis of muscle cross-sectional area

Isopentane-frozen muscles were prepared for sectioning in a cryostat at −20°C to −30°C. Sections of 8–10 μm were obtained and stained using the hematoxylin and eosin (H&E) staining method. Stained samples were subsequently imaged by light microscopy. Muscle cross-sectional area was determined using ImageJ software to manually trace the circumference of individual fibers (Schneider et al, 2012).

## Immunoblotting

Protein from C2C12 cells were extracted using a mammalian lysis buffer (50 mM HEPES pH 7.0, 150 mM NaCl, 10% glycerol, 1% Triton X-100, 10 mM sodium pyrophosphate, 100 mM NaF, 1 mM EGTA, 1.5 mM $MgCl_2$). Immunoblotting analyses were performed using soluble fraction of whole protein lysates run on 7.5–12% acrylamide gels. Proteins were blotted on to nitrocellulose membranes, blocked with either 10% skim milk or 5% BSA in TRIS-buffered saline, 0.1% Tween (TBS-T) for 30–60 min, and probed with antibodies against iNOS (1:5,000 – 1:3,000), and pAMPK (1:5,000 – 1:1,000), AMPK (1:5000 – 1:1,000), pACC (1:5000–1:1,000), ACC (1:1,000), pS6K (1:2,000), S6K (1:5,000), pS6 (1:5,000), S6 (1:1,000), and α-tubulin (1:5,000). After washing with TBS-T, blots were subsequently probed with the appropriate horseradish peroxidase-conjugated secondary antibodies (1:5,000) and exposed to ECL reagent (Western Lightning Plus, Perkin Elmer). Signal was determined by exposure to photosensitive films. Quantifications of Western blot band signal density were performed using ImageJ software (Schneider et al, 2012).

## Immunofluorescence and myotube diameter measurements

For immunofluorescence experiments, cells were first fixed in 3% paraformaldehyde for 30 min. Cells were then permeabilized in a solution containing 0.1% Triton X-100 and 1% goat serum in phosphate-buffered saline. After washing with 1% goat serum phosphate-buffered saline, cells were incubated with antibodies against myosin heavy chain (MF-20; 1:1,000) and myoglobin (1:500) to visualize myotubes for 1 h at room temperature. Following further washing, cells were then incubated with appropriately labeled Alexa Fluor® (Invitrogen) secondary antibodies (1:500) for one hour at room temperature. Nuclei were visualized by DAPI staining (Miron et al, 2004). Cells were visualized using a Zeiss Observer.Z1 microscope with a 40× oil objective, and images were obtained using an AxioCam MRm digital camera. Myotube diameters were measured at 3 points along each cell using AxioVision Rel. 4.8 or Carl Zeiss Zen2 (blue) software.

## Metabolic assays

Cellular glucose consumption and lactate production rates were determined by measuring media concentrations of glucose and lactate with a Flux Bioanalyzer (NOVA Biomedical) as previously described (Vincent et al, 2015). Extracellular flux rates were measured using the Seahorse Bioscience XFe24 Analyzer (Seahorse Bioscience—Agilent Technologies). On the day of seeding, the wells of an XF24 culture plate were coated with 1% gelatin for 30 min at room temperature. Excess gelatin was removed and C2C12 myoblasts were seeded at $2 \times 10^4$ cells/well in 20% FBS, 1% penicillin, streptomycin-supplemented DMEM. The next day, cells were switched to 2% HS, 1% penicillin/streptomycin-supplemented DMEM to initiate differentiation. On day 3 or day 4 of differentiation, cells were treated with IFNγ (100 U/ml)/TNFα (20 ng/ml) and/or AICAR or metformin (0.5 mM) in 2% HS, 1% pen./strep. media for 24 h. Approximately 1 h before beginning the assay, cells were switched to Seahorse Assay Media (Agilent 102353-100) supplemented with 25 mM glucose (Sigma—G7528) and 2 mM glutamine (ThermoFisher—25030149), adjusted to pH 7.4 at 37°C. Cells were subsequently incubated in a non-$CO_2$ incubator for 1 h. The XF assay consisted of an initial 3 cycles of mix (3 min), pause (2 min), measure (3 min) to establish basal respiration. Oligomycin (final contraction 1 μM), FCCP (final concentration 1.5 μM), and rotenone and antimycin A (final concentrations 1 μM) were injected sequentially, with two measurement cycles (see above) following each injection. After the Seahorse assay was complete, cellular protein content was measured using an SRB assay as described in (Vichai & Kirtikara, 2006). In brief, cells were fixed in with trichloroacetic acid (TCA) at a final concentration of 3.3% (wt/vol) for 1 h at 4°C. After fixation, cells were washed with running distilled water and dried at room temperature. Fixed cells were stained with a 1% acetic acid solution containing 0.057% SRB (wt/vol) for 30 min. at room temperature, followed by four washes with 1% acetic acid. Staining was solubilized by the addition of 10 mM Tris, pH 10.5, and absorbance at 510 nm was read using a Synergy Mx MultiMode Plate Reader. Absorbance at 510 nm was then relativized to the average absorbance for non-treated cells to determine the relative protein units (RPU). Extracellular flux rates were then normalized to the RPU for each well.

**The paper explained**

**Problem**

Cachexia is a debilitating condition of muscle atrophy that arises in patients with chronic inflammatory diseases (e.g., cancer and sepsis). The loss of muscle mass is associated with a dramatic decrease in quality of life, reduced efficacy of treatment of the overlying condition, and increased risk of mortality. However, to date, only palliative treatment options are available for cachectic patients. Development of effective therapies requires the identification of druggable molecular targets, but few candidates have been identified for anti-cachectic therapy. Given the inflammatory and metabolic nature of cachexia, we hypothesized that activators of AMP-activated protein kinase (AMPK) could be beneficial due to its anti-inflammatory and metabolic homeostatic properties. However, AMPK has also been implicated in muscle atrophy, and so it was unclear whether AMPK activators would be helpful or detrimental for cachexia-associated muscle wasting.

**Results**

We found that the AMPK activators AICAR and A-769662 were effective at preventing atrophy in cultured myotubes treated with inflammatory cytokines. Also, AICAR was found to reduce muscle loss in cancer and septic models of cachexia. The effects of AICAR and A-769662 were found to be synergistic, and the effects of AICAR were blocked by an AMPK inhibitor, suggesting that these compounds act through AMPK. In contrast, the anti-diabetic drug metformin, which has also been described as an AMPK activator, had no beneficial effects. Instead, both cytokines and metformin were found to inhibit oxidative respiration and promote glycolytic metabolism and suppressed anabolic signaling.

**Impact**

This study demonstrates that AMPK activators could represent a novel avenue for the development of anti-cachectic therapies. Use of AMPK activators will likely be particularly beneficial for cachexia treatment in diseases for which AMPK activators are also being explored as a treatment option for the primary disease, such as in cancer. However, the efficacy of AMPK activation is likely to depend on the mechanism of activation, as activation of AMPK indirectly through mitochondrial inhibition was not associated with any beneficial effects.

### De novo protein synthesis

Analysis of de novo protein synthesis rates was performed as previously described (Di Marco et al, 2012). In brief, C2C12 myotubes were treated with IFNγ/TNFα and AMPK agonists, as described above. At the indicated time points, cells were incubated with Easy-Tag™ Express Protein Labelling Mix (PerkinElmer) at approximately 27.5 μCi/ml radioactivity for 30 min at 37°C. After, cells were lysed directly in SDS dye, resolved on a polyacrylamide gel, and stained with Coomassie Blue. Radioactivity was visualized by exposing dried gels to photosensitive film.

### RT–qPCR

Total RNA was extracted from differentiated C2C12 cells treated as described above using TRIzol® (Invitrogen) according to the manufacturer's instructions. RNA quantity and quality were determined by spectrophotometric analysis using a Thermofisher NanoDrop™ reader (ND-1000) and by agarose gel electrophoresis. A reverse transcription reaction was performed using M-MLV Reverse

Transcriptase (New England Biolabs). A negative control lacking the reverse transcriptase was also performed. Generated cDNA was then analyzed by qPCR using a Corbett RG-6000, and levels were normalized to GAPDH and subsequently relativized to the non-treated control for all experimental replicates. Primer sequences are given in Appendix Table S1.

**AMPK siRNA knockdown and overexpression**

C2C12 myoblast cells were transfected using jetPEI (Polyplus Transfection) according to the manufacturer's instructions. For myoblast knockdown experiments, cells were transfected at ~50% confluence with 50 nM of siRNA targeting the AMPKα1 subunit (Origene: #SR417172) or a universal scrambled negative control siRNA (Origene: #SR30004). 24 h after transfection, cells were treated with cytokines, as described above. 24 h after treatment, cells were collected and protein extracts were collected. For myoblast AMPK overexpression experiments, myoblasts were transfected as above with either a pcDNA3 vector expressing the cDNA sequence of the human AMPKα1 subunit or an empty vector control.

**Statistics and data processing**

For non-animal studies, $n$ values indicate the number of times the experiment was independently replicated. For animal studies, $n$ values indicate the number of animals per cohort. For animal studies, sample sizes were based on previous studies and expertise, with a minimum of three mice for any given statistical analysis. For *in vitro* studies, experimental replicates were excluded if the negative (non-treated) and positive (cytokine-treated) samples did not show an appropriate inflammatory response. For *in vivo* studies, mice were excluded if they developed complications unrelated to the cachexia phenotype or if they developed humane-intervention end-point complications early in the course of the study (e.g., ulceration of the tumor mass). Samples for measuring myotube widths and muscle fiber cross-sectional areas were blinded before acquisition of images and during quantification. Graphs represent the mean, with error bars showing either the standard error of the mean for biological replicates or the standard deviation for technical replicates. Significance between means was first determined using ANOVA with Brown-Forsythe and D'Agostino & Person tests for variance similarity and normal distribution, respectively. Significance *P*-values were computed using Fisher's uncorrected LSD or the Student's *t*-test in GraphPad Prism version 6 and 7 for Windows, GraphPad Software, La Jolla California USA, www.graphpad. All *P*-values for main figures and tables and extended view figures can be found in Appendix Tables S2–S14.

**Data availability**

All relevant data pertaining to the studies conducted within this manuscript are available from the authors upon request.

**Expanded View** for this article is available online.

## Acknowledgements
We are grateful to Erzsebet Nagy Kovacs for assistance with the animal studies. We thank Dr. John Silvius and Dr. Gerardo Ferbeyre for their ideas, suggestions, and guidance. This work is funded by a Canadian Cancer Society Research Institute (CCSRI) Innovation Grant (702565), a CIHR operating grant (MOP-142399, PJT-156397), and a Qatar National Research Fund (QNRF) (NPRP8-457-3-101) to I.E.G, as well as a CIHR operating grant (MOP-93799) to R.G.J., Canadian Cancer Society Research Institute – Innovation grant (#703394) to S.S.W. G.R.S. is a Canada Research Chair in Metabolism and Obesity and the J. Bruce Duncan Chair in Metabolic Diseases and acknowledges funding from CIHR-125980-1 and the Canadian Cancer Society Research and Innovation Fund. D.T.H. was funded by a scholarship received from the Canadian Institute of Health Research (CIHR) funded Chemical Biology Program at McGill University.

## Author contributions
DTH contributed to conceptualization, conducted the investigation and validation of experimental findings, wrote the original draft, and performed the formal analysis and visualization of experimental findings. TG assisted with investigations, validations, and formal analysis of the metabolic studies, including the GC-MS and oxidative respiration studies. JFM, BJS, JS, AMKT, SM, and AO contributed to the investigations and validations of the cachectic models as well as to help with experiments to address reviewers' comments. RJF assisted in the conceptualization of animal and discussed results. NB helped with the design and execution of animal study. AP contributed with reagent, the interpretation of the results, and reviewed and edited the manuscript. SSW helped interpret the results, contributed to the design of animal studies, and reviewed and helped edit the manuscript. SDM assisted with conceptualization, data analysis, and helped edit and review the manuscript. GRS helped in the conceptualization of animal studies, discussed and helped in the analysis of the results, and reviewed and helped edit the manuscript. RGJ provided materials, equipment, and supervision for conducting the metabolic studies and discussed and helped the analysis of the results and reviewed and helped edit the manuscript. I-EG conceptualized, established, and directed the execution of research goals, interpreted the data, reviewed, and edited the manuscript.

## Conflict of interest
The authors declare that they have no conflict of interest.

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
