## [Review Process File · EMBO Molecular Medicine]

The AMPK agonist 5-Aminoimidazole-4-carboxamide ribonucleotide (AICAR), but not metformin, prevents inflammation-associated cachectic muscle wasting

Derek T. Hall, Takla Griss, Jennifer F. Ma, Brenda Janice Sanchez, Jason Sadek, Anne Marie K. Tremblay, Souad Mubaid, Amr Omer, Rebecca J. Ford, Nathalie Bedard, Arnim Pause, Simon S. Wing, Sergio Di Marco, Gregory R. Steinberg, Russell G. Jones, and Imed-Eddine Gallouzi

Review timeline:

Submission date:	21 July 2017
Editorial Decision:	26 September 2017
Revision received:	07 March 2018
Editorial Decision:	16 April 2018
Revision received:	01 May 2018
Accepted:	04 May 2018

Editor: Roberto Buccione/Céline Carret

Transaction Report:

1st Editorial Decision

26 September 2017

Thank you for the submission of your manuscript to EMBO Molecular Medicine.

I apologise for the very unusual delay in getting back to you on your manuscript. In fact, we experienced significant difficulties in securing expert and willing reviewers, and then obtaining their evaluations in a timely fashion, in part due to the overlapping holiday season. Furthermore, additional internal discussion was required to reach final decision.

As you will see, although the three evaluations are fundamentally non-negative, reviewers 1 and 2 are more reserved and raise important concerns. These are in part overlapping and impinge on the mechanistic basis for some of the observations. Namely, the mechanistic link between AICAR/metformin/AMPK is weak, and the evidence of efficacy of AICAR in C26 and as a treatment for cancer cachexia as presented is not strong, which reduces the translational impact of the study (a very important aspect for our journal). Reviewer 1 suggests that perhaps AICAR treatment would be more effective in a less dramatic model of muscle wasting than the C26, which is so severe that differences are harder to detect. Reviewer 2 also notes that it would be important to understand how AMPK is perturbing the known mechanisms underlying TNF-alpha/IFN-gamma induced atrophy. Indeed, the fact that TNF/Interferon treatment on its own activates AMPK, would seem to invalidate the entire point of furthering activating AMPK via AICAR.

These issues were discussed during our cross-commenting exercise and as mentioned above, internally. It is appreciated that the paradoxical activation of AMPK by TNF/INF, which is proposed to be pathological, versus AICAR-mediated activation which is protective, is a central item of novelty here. However, there is agreement, including from reviewer 3, that the mechanistic basis of the difference is not convincingly shown and the therapeutic application of

AICAR underwhelming.

In conclusion, while publication of the paper cannot be considered at this stage, we are willing to consider a substantially revised manuscript, addressing the reviewers' concerns as mentioned above including with further experimentation where required.

Please note that it is EMBO Molecular Medicine policy to allow a single round of revision only and that, therefore, acceptance or rejection of the manuscript will depend on the completeness of your responses included in the next, final version of the manuscript.

As you know, EMBO Molecular Medicine has a "scooping protection" policy, whereby similar findings that are published by others during review or revision are not a criterion for rejection. However, I do ask you to get in touch with us after three months if you have not completed your revision, to update us on the status. Please also contact us as soon as possible if similar work is published elsewhere.

EMBO Molecular Medicine now requires a complete author checklist (<http://embomolmed.embopress.org/authorguide#editorial3>) to be submitted with all revised manuscripts. Provision of the author checklist is mandatory at revision stage; The checklist is designed to enhance and standardize reporting of key information in research papers and to support reanalysis and repetition of experiments by the community. The list covers key information for figure panels and captions and focuses on statistics, the reporting of reagents, animal models and human subject-derived data, as well as guidance to optimise data accessibility.

Please carefully adhere to our guidelines for authors (<http://embomolmed.embopress.org/authorguide>) to accelerate manuscript processing in case of acceptance.

I look forward to receiving your revised manuscript in due time.

Should you find that the requested revisions are not feasible within the constraints outlined here and choose, therefore, to submit your paper elsewhere, we would welcome a message to this effect.

***** Reviewer's comments *****

Referee #1 (Comments on Novelty/Model System for Author):

The manuscript by Hall et al. addresses the important question of mechanisms of muscle wasting in conditions of inflammation, the potential roles of AMPK in cytokine-induced wasting, and the mechanisms and potential utility of modulating AMPK for preventing muscle wasting in cancer cachexia. Overall the strengths of the manuscript are: it is well written, well researched and grounded in the literature, which is well-referenced; the subject matter is of great interest to a wide community of muscle biologists and cachexia and cancer researchers; the data showing different effects of AICAR versus metformin on myotube wasting despite similar activation of AMPK are strong, novel and important; the demonstration of myotube-sparing effects of the AMPK agonist A-769662 suggests this compound could have utility for preventing muscle loss; the data in the C26 cachexia model validates activity for AICAR and metformin as anti-cancer agents, and suggests potential for AICAR as an anti-cachexia agent. Thus overall there are important contributions made, particularly the distinction between metformin and AICAR effects despite their apparent common activation of AMPK when using pAMPK and pACC as readouts. Given that metformin is in clinical trials for cytokine-associated muscle wasting conditions, the demonstration that its Complex I inhibitory effects might take precedence over its AMPK activating effects is important. Moreover, this distinction has implications beyond the fields of muscle, cachexia and cancer.

There are weaknesses in the study, however, that reduce its overall impact. The mechanistic link between AICAR/metformin/AMPK is somewhat weak, and the evidence of efficacy of AICAR in C26 and as a treatment for cancer cachexia as presented is not strong. In my opinion, at least one of these two needs to be strengthened to make this manuscript high impact.

Referee #1 (Remarks for Author):

There are weaknesses in the study, however, that reduce its overall impact. The mechanistic link between AICAR/metformin/AMPK is somewhat weak, and the evidence of efficacy of AICAR in C26 and as a treatment for cancer cachexia as presented is not strong. In my opinion, at least one of these two needs to be strengthened to make this manuscript high impact.

1. Regarding mechanism, the studies showing differential effects of AICAR versus metformin are done in C2C12 myotubes treated with cytokines, while the knockdown study demonstrating that the effects are AMPK dependent are in myoblasts. Controls are missing in those myoblast experiments and the readouts are necessarily indirect when there are no myotubes to measure. This weakens the link the authors try to forge between AICAR and AMPK. The Compound C studies are not definitive either, because CC seems to have a hypertrophic effect on the myotubes at baseline and its effects on glucose/lactate are small, blocking at most 50% of the AICAR effect, although effects on pS6 and iNOS are much more pronounced. Overall, if the mechanism cannot be addressed more specifically (e.g. by genetic knockdown of AMPK in myotubes through viral infection of shRNA), then the mechanistic aspect of the study still requires more investigation. In that case, the authors should soften the language around the effects of AICAR being mediated by direct activation of AMPK because all the evidence is indirect and not exceptionally strong.

As well, the use of A-769662 is presented as a second validation of AMPK activation inhibiting cytokine-induced myotube wasting because it is reported to target a different subunit. Indeed, the data show reduced myotube wasting, but there is a missed opportunity to show that these effects might be additive or synergistic with AICAR, or that it might be effective *in vivo*.

2. The *in vivo* study which claims to show less muscle wasting with AICAR in C26 cachexia is not convincing as presented, but that might be rectified by more careful presentation of the existing data. A large number of mice were used in the *in vivo* study, a strength, but the data presentation is curious and includes only a subset of these mice. There is not much difference in weight loss between the AICAR and other groups, although that could be better highlighted with a bar graph of tumor-free body weight. Also, I suggest that you report the muscle mass (and all of them-it is not clear why 13 mice were used but a different number of TA and G muscles are reported in the muscle graph) as a fraction of the starting body weight rather than as percentage muscle loss. This will account for differences in mouse size at the start and will permit comparison of groups. It is not clear how a percentage weight loss could be calculated anyway because the muscle mass at the start was unknown. As well, you saw increased tumor growth but reduced weight loss in the AICAR group, suggesting an anti-cachexia effect in this setting. I would suggest plotting percentage weight loss or muscle mass versus tumor mass to see whether you have dissociated these two conditions.

The cross-sectional area data were derived from only 3 mice per point and are not consistent with the fairly dramatic loss of muscle mass in the AICAR group (20%). Why this disconnect?

The data here are not sufficiently strong to warrant clinical trials for AICAR in cachexia, for example, and so the overall impact is not strong. The model is not pre-clinical, but only experimental and the magnitude of the response is not large. However, I suspect that AICAR treatment would be more effective in a less dramatic model of muscle wasting than the C26, which is so severe that differences are harder to detect.

Other points:

1. When discussing effects in C2C12 cultures, please use "myotube atrophy" and "myotubes" rather than "muscle wasting" or "muscle".
2. If the glucose/lactate data are not normalized to protein content, is there a difference? What is changing, the numerator or the denominator? These are done in conditions of atrophy, so there is less protein. Does this really reflect different metabolism?
3. Please report the siRNA sequence and the scrambled control sequence or catalog number so these studies can be reproduced by others.
4. Please provide more clarity on the clinical relevance of these doses of drugs. Is it possible to get the in vivo levels of AICAR to 500mM or deliver 500mg/kg/day in humans?
5. What is the detection method for the WBs? ECL?
6. How/was the quality/quantity of RNA for qPCR assayed? How were the results normalized?
7. How many mice were excluded? Were they randomized in their cages or were all mice in one treatment group in the same cage?
8. Figure 1b is too small. I suggest expanding it to the full width of the figure so that the differences in the cultures can be appreciated.

Referee #2 (Remarks for Author):

The most direct published mechanisms underlying TNF-alpha/IFNgamma induced atrophy are:

- 1) Activation of NF-kappaB signaling, which increases MuRF1 levels, which directly causes Myosin heavy chain breakdown. (Cai et al, Cell. 2004 Oct 15;119(2):285-98.)
- 2) Stimulation of ActivinA levels in muscle (shown with other cytokines, albeit not TNF-alpha): (for example, Matsuyama; Int J Cancer. 2015 Dec ; cause and effect shown in: Trendelenburg et al Skelet Muscle. 2012 Feb 7;2(1):3.)

It would be important to understand how AMPK is perturbing these mechanisms. Therefore it seems reasonable to request that the authors look at MuRF1 and MAFbx/Atrogin-1 upregulation in vivo +/- AICAR or the more specific AMPK activator. Also, looking at SMAD2/3 levels would be useful.

The authors point out that AMPK activation could be helpful under inflammatory settings in particular, thus looking at NF-kappaB activation (perhaps by checking IkappaB breakdown and MuRF1 upregulation) would be indicated.

As the authors point out, there are now more specific AMPK activators available, so use of those are more convincing. See for example Myers et al, Science, July 2017.

Specific points:

Figure 1A: pACC seems activated in lane 4 (just with IFNgamma/TNFalpha) treatment... this is shown in the quantification as well - thus it seems like cytokine treatment is sufficient to activate AMPK, which would argue against the entire thesis of the paper (that AMPK activation blocks cytokine effects). There' no real difference in pACC with AICAR above IFNgamma/TNFalpha. The authors make note of this, but don't seem to recognize that this undercuts the entire paper - what's the point of adding an AMPK activator to counter cytokine induced atrophy, if the cytokines are already activating AMPK, and causing atrophy? (Same

point re pAMPK)

Figure 3: the relative pS6/S6 determination is a bit misleading, given that the cytokines are dramatically decreasing S6 levels. Also, there is an inconsistency in total S6 across the lanes, making this experiment difficult to interpret. This should be repeated, along with phospho-p70S6K/total p70 determination, along with the whole blot being shown.

Figure 4: Decreasing total levels of iNOS was not the prior mechanism leading to atrophy - rather, releasing iNOS from the dystroglycan complex, leading to decreased Akt phosphorylation. It would be surprising if AMPK was perturbing that mechanism... so how do the authors think iNOS is functioning here?

Figure 7: What is the effect of AICAR alone on muscle mass? Here too, examination of the E3 ligases would seem to be essential. (MuRF1 and MAFbx/Atrogin-1_

Referee #3 (Comments on Novelty/Model System for Author):

C2C12 myotubes treated with IFN γ /TNF α are well accepted in vitro model of cytokine induced muscle wasting; intra-muscular injection of IFN γ /TNF α cytokines in mice provides a tumor-free in vivo model of muscle wasting; C26 adenocarcinoma tumor-bearing mice are in vivo cachectic model of muscle wasting.

Referee #3 (Remarks for Author):

The paper of Hall et al is an interesting and elegant study, which identifies a novel protective function of AMPK against cytokine-driven atrophy. By using in vitro and in vivo models of cytokine-driven muscle wasting, the authors have shown that the AMPK agonist AICAR suppresses cytokine-induced atrophy. Prevention of atrophy was associated with reduction of glycolytic flux, restoration of oxidative metabolism and suppression of iNOS/NO pathway in the context of IFN γ /TNF α treatment. Compelling evidence indicates that the protective role of AICAR is mediated by AMPK and is linked to its anti-inflammatory properties, being independent from its well-known role in suppression of anabolism upon metabolic stress. Indeed, the reduction of mTOR signaling and protein synthesis, induced by IFN γ /TNF α , was blunted by AICAR concomitant stimulation. In line with this, the ability of AICAR treatment to restore mTOR signaling was impaired in the presence of Compound C, a specific AMPK inhibitor. This is a phenomenon that deserves further exploration and the present report contributes to that effort. The manuscript is technically sound and the results are very convincing. The experiments are carefully performed and the results are clearly presented. I would recommend publication of this paper essentially as it is.

Minor observations:

Line 118: "Fig. 1A"

Lines 131-132: rephrase "AICAR treatment, but not metformin trended to higher mRNA levels fo MyoD (significant) and myogenin"

Fig. 1A: the right panel lacks of p values on the bars

Line 317: "... sufficient to induce cachexia"

Referee #1 (Comments on Novelty/Model System for Author):

“The manuscript by Hall et al. addresses the important question of mechanisms of muscle wasting in conditions of inflammation, the potential roles of AMPK in cytokine-induced wasting, and the mechanisms and potential utility of modulating AMPK for preventing muscle wasting in cancer cachexia. Overall the strengths of the manuscript are: it is well written, well researched and grounded in the literature, which is well-referenced; the subject matter is of great interest to a wide community of muscle biologists and cachexia and cancer researchers; the data showing different effects of AICAR versus metformin on myotube wasting despite similar activation of AMPK are strong, novel and important; the demonstration of myotube-sparing effects of the AMPK agonist A-769662 suggests this compound could have utility for preventing muscle loss; the data in the C26 cachexia model validates activity for AICAR and metformin as anti-cancer agents, and suggests potential for AICAR as an anti-cachexia agent. Thus overall there are important contributions made, particularly the distinction between metformin and AICAR effects despite their apparent common activation of AMPK when using pAMPK and pACC as readouts. Given that metformin is in clinical trials for cytokine-associated muscle wasting conditions, the demonstration that its Complex I inhibitory effects might take precedence over its AMPK activating effects is important. Moreover, this distinction has implications beyond the fields of muscle, cachexia and cancer”.

“There are weaknesses in the study, however, that reduce its overall impact. The mechanistic link between AICAR/metformin/AMPK is somewhat weak, and the evidence of efficacy of AICAR in C26 and as a treatment for cancer cachexia as presented is not strong. In my opinion, at least one of these two needs to be strengthened to make this manuscript high impact”.

Referee #1 (Remarks for Author):

“There are weaknesses in the study, however, that reduce its overall impact. The mechanistic link between AICAR/metformin/AMPK is somewhat weak, and the evidence of efficacy of AICAR in C26 and as a treatment for cancer cachexia as presented is not strong. In my opinion, at least one of these two needs to be strengthened to make this manuscript high impact”.

1. “Regarding mechanism, the studies showing differential effects of AICAR versus metformin are done in C2C12 myotubes treated with cytokines, while the knockdown study demonstrating that the effects are AMPK dependent are in myoblasts. Controls are missing in those myoblast experiments and the readouts are necessarily indirect when there are no myotubes to measure”.

We apologize for the lack of clarity on our part while describing this section. The purpose of the knockdown study in myoblasts was not to demonstrate the AMPK dependency of all aspects of the effect of AICAR on cytokine-induced atrophy. It is only used to show that the down regulation of iNOS, which is only expressed when cells are treated with cytokines, is dependent on AMPK. Therefore, non-cytokine treated controls cannot be tested, as they do not express iNOS. We have altered the description of this section (lines 262-267) to make it clear that we are only assessing effects on iNOS expression in myoblasts, not the full range of the cachectic phenotype.

- “This weakens the link the authors try to forge between AICAR and AMPK. The Compound C studies are not definitive either, because CC seems to have a hypertrophic effect on the myotubes at baseline and its effects on glucose/lactate are small, blocking at most 50% of the AICAR effect, although effects on pS6 and iNOS are much more pronounced. Overall, if the mechanism cannot be addressed more specifically (e.g. by genetic knockdown of AMPK in myotubes through viral infection of shRNA), then the mechanistic aspect of the study still requires more investigation. In that case, the authors should soften the language around the effects of AICAR being mediated by direct activation of AMPK because all the evidence is indirect and not exceptionally strong. As well, the use of A-769662 is presented as a second validation of AMPK activation inhibiting cytokine-induced myotube wasting because it is reported to target a different subunit. Indeed, the data show reduced myotube wasting, but there is a missed opportunity to show that these effects might be additive or synergistic with AICAR, or that it might be effective in vivo”.

We thank the reviewer for this suggestion, and we have conducted synergy experiments with AICAR and A-769662. We show that AICAR and A-769662, when treated at sub-effective doses, can synergistically prevent the effects of cytokine-induced myotube wasting (Fig. 7). Therefore, we

thank again the reviewer for this great suggestion, and in our opinion, this experiment strengthens the idea that AICAR, and A-769662, prevent cytokine-induced myotube atrophy through AMPK.

In addition, we have adjusted the text of the manuscript throughout to emphasize the differential effects of AICAR/A-769662 and metformin/cytokines on metabolism as a potential explanation for their differential effects and removed references to direct or indirect AMPK activation. We have also lowered the tone to state that our data suggest, but does not prove, that the effects we observed for AICAR/A-769662 are mediated through AMPK.

2. “The in vivo study which claims to show less muscle wasting with AICAR in C26 cachexia is not convincing as presented, but that might be rectified by more careful presentation of the existing data. A large number of mice were used in the in vivo study, a strength, but the data presentation is curious and includes only a subset of these mice. There is not much difference in weight loss between the AICAR and other groups, although that could be better highlighted with a bar graph of tumor-free body weight [...]. It is not clear how a percentage weight loss could be calculated anyway because the muscle mass at the start was unknown”.

We have significantly adjusted the presentation of the mouse model data to address the concerns about clarity as raised by the reviewer.

The presentation of muscle mass as percent wasting was calculated as percentage of the average of the saline treated cohort, but this was not clear in our original submission. Therefore, in order to improve the clarity, we have elected to present our body and tissue weight data as a table instead of graphs (Table 1), which is an accepted presentation method for cachexia datasets (PMID: 15286803, 21949739). In this format, the difference in muscle weight and percent body weight change between the C26 and C26 + AICAR cohorts can be more easily appreciated.

- “Also, I suggest that you report the muscle mass (and all of them-it is not clear why 13 mice were used but a different number of TA and G muscles are reported in the muscle graph) as a fraction of the starting body weight rather than as percentage muscle loss. This will account for differences in mouse size at the start and will permit comparison of groups”.

The discrepancy between number of TA and gastrocnemius muscle was due to the inclusion of an initial pilot study in which only the gastrocnemius was collected. To prevent confusion and to provide a more rigorous and consistent methodology, we have elected to remove the results of the pilot studies as the full data-set was not collected from them. Removal of these mice does not significantly affect the trends or effect sizes of our results.

We purchased our mice to be age matched and weighing on average 23g, to allow for comparisons between groups, and have added text to the methods section to indicate this.

- “As well, you saw increased tumor growth but reduced weight loss in the AICAR group, suggesting an anti-cachexia effect in this setting. I would suggest plotting percentage weight loss or muscle mass versus tumor mass to see whether you have dissociated these two conditions”.

As suggested by the reviewer, we have plotted the muscle mass against the tumor burden and observed that there is a loss of correlation between tumor burden and muscle mass loss in the C26 + AICAR, but not the C26 or C26 + metformin, cohorts (New Fig. S6). This suggested to us that AICAR was preventing the effect of increasing tumor growth on muscle loss in the later stages of the model. To test this, we assessed the muscle mass of mice at day 14 and day 21 post-C26 inoculation in which AICAR treatment was begun at day 12, as in our main study. We observed, as previously demonstrated by our laboratory as well as others (PMID: 15286803, 22692539), that a significant portion of the wasting (~50%) had already occurred by day 14 and was unaffected by the two days of AICAR treatment. In contrast, AICAR treated mice loss minimal muscle mass in the following 7 days, from day 14 to day 21, suggesting that AICAR treatment was able to block the additional 50% of wasting that occurs after day 14. Therefore, we surmise that the AICAR efficacy in our model is limited by the need to treat later in disease progression so as to not affect tumor

burden and subsequently muscle atrophy indirectly, as was demonstrated when our treatment with AICAR was started at day 9 (Fig. S4).

- “The cross-sectional area data were derived from only 3 mice per point and are not consistent with the fairly dramatic loss of muscle mass in the AICAR group (20%). Why this disconnect?”

We thank the reviewer for observing the discrepancy between our weight and CSA data, something we had not previously noted. Further review of the data discovered that the CSA graph had unintentionally been presented as total number of fibers counted for each range, not relative percentage. We have fixed this in the new manuscript. In addition, we have counted fibers from an additional mouse for each cohort, bringing the total number of mice assessed to four for each. The CSA data now clearly shows that there is a decrease in CSA in the C26 + AICAR cohort when compared to saline, but that it is also recovered when compared to the C26 cohort, reflecting the results seen in our muscle weights (Fig. 8).

“The data here are not sufficiently strong to warrant clinical trials for AICAR in cachexia, for example, and so the overall impact is not strong. The model is not pre-clinical, but only experimental and the magnitude of the response is not large. However, I suspect that AICAR treatment would be more effective in a less dramatic model of muscle wasting than the C26, which is so severe that differences are harder to detect”.

As per the reviewer’s suggestion, we have tested the efficacy of AICAR in a less severe model of cachexia. We elected to try a model of sepsis associated cachexia to see if the beneficial effects of AMPK activators extended to other forms of inflammatory associated muscle wasting. In a model of LPS-induced muscle wasting, AICAR, but not metformin, completely prevented the loss of muscle mass (see new Fig. 9). Therefore, in a septic model of cachexia, we have demonstrated a proof-of-principle that AMPK activators like AICAR can effectively protect muscle tissue from atrophy.

Taken together, our new results more clearly present the prevention of muscle wasting by AICAR in the C26 cancer model of cachexia, demonstrate that the efficacy of AICAR in this model is strong, but limited by the necessity to treat later to avoid affecting tumor growth, and that AICAR is also effective in LPS induced cachexia. Therefore, we feel that we have significantly strengthened the evidence that AICAR, or AMPK activating compounds like AICAR, could be effective as anti-cachectic agents in a variety of contexts.

“Other points:

1. “When discussing effects in C2C12 cultures, please use "myotube atrophy" and "myotubes" rather than "muscle wasting" or "muscle"”.

We have corrected the text to refer to myotubes when C2C12 cultures were used.

2. “If the glucose/lactate data are not normalized to protein content, is there a difference? What is changing, the numerator or the denominator? These are done in conditions of atrophy, so there is less protein. Does this really reflect different metabolism?”

In the IFN γ /TNF α model of wasting in C2C12, there is no significant wasting that occurs at the time points when the metabolic assays were conducted. Indeed, we have seen that, at 24h when the cells were assayed for metabolic function, the inhibition of S6 phosphorylation induced by cytokines has not yet occurred and there is no observable effect on myotube size. Therefore, since there is no significant effect on protein content at the 24h time point, the trends observed in the raw glucose/lactate are unaffected by protein normalization.

3. “Please report the siRNA sequence and the scrambled control sequence or catalog number so these studies can be reproduced by others”.

The catalog number has been provided in the methods section.

4. “Please provide more clarity on the clinical relevance of these doses of drugs. Is it possible to get the in vivo levels of AICAR to 500mM or deliver 500mg/kg/day in humans?”

We have included a statement on the clinical relevance of the doses in the methods section.

5. “What is the detection method for the WBs? ECL?”

The detection method is ECL, and we have indicated this in our methods section.

6. “How/was the quality/quantity of RNA for qPCR assayed? How were the results normalized?”

The quality and quantity was assessed with a Nanodrop spectrometer and by agarose gel electrophoresis. The results were normalized to GAPDH. This information has been included in our methods section.

7. “How many mice were excluded? Were they randomized in their cages or were all mice in one treatment group in the same cage?”

Mice were randomized to treatment groups by cage. As described above (remark #2), we have excluded mice from a pilot study in which not all of the muscle tissues were collected, which did not significantly affect the outcome of the study.

8. “Figure 1b is too small. I suggest expanding it to the full width of the figure so that the differences in the cultures can be appreciated”.

The panel has been expanded to the full width of the figure, as requested.

Referee #2 (Remarks for Author):

“The most direct published mechanisms underlying TNF-alpha/IFNgamma induced atrophy are:

1) Activation of NF-kappaB signaling, which increases MuRF1 levels, which directly causes Myosin heavy chain breakdown. (Cai et al, Cell. 2004 Oct 15;119(2):285-98.)

2) Stimulation of ActivinA levels in muscle (shown with other cytokines, albeit not TNF-alpha): (for example, Matsuyama; Int J Cancer. 2015 Dec ; cause and effect shown in: Trendelenburg et al Skelet Muscle. 2012 Feb 7;2(1):3.)”

“It would be important to understand how AMPK is perturbing these mechanisms. Therefore it seems reasonable to request that the authors look at MuRF1 and MAFbx/Atrogin-1 upregulation in vivo +/- AICAR or the more specific AMPK activator. Also, looking at SMAD2/3 levels would be useful”.

We have included data on the upregulation of MuRF1 and Atrogin-1/MAFbx. In both the C26 and LPS models (see above) there is a significant upregulation of the mRNA of these E3 ligases in the presence of C26 tumors or after the injection of LPS (Fig. 8, 9). AICAR treatment reduced the expression of Atrogin-1/MAFbx in both models (Fig. 8, 9). MuRF1 was less affected by AICAR in the C26 model, but was significantly reduced in the LPS model (Fig. 8, 9).

We looked at SMAD2/3 levels in the cytokine model in C2C12, but observed no obvious effects of either cytokine or AICAR or Metformin treatment. Therefore, we did not feel it was necessary to include these results in our manuscript.

“The authors point out that AMPK activation could be helpful under inflammatory settings in particular, thus looking at NF-kappaB activation (perhaps by checking IkappaB breakdown and MuRF1 upregulation) would be indicated”.

As indicated above, we assessed MuRF1 expression in our models. We saw a trend towards reduced expression in the C26 model and a significant reduction in the LPS model.

“As the authors point out, there are now more specific AMPK activators available, so use of those are more convincing. See for example Myers et al, Science, July 2017”.

At the time our study was conducted, A-769662 was the most specific AMPK activator commercially available that we were aware of. As described in the paper, we used A-769662 as a more specific AMPK activator and confirmed the effects of AICAR. We have also included new data (detailed above in our answers to reviewer 1) showing that A-769662 and AICAR can synergistically affect cytokine-driven myotube atrophy.

“Specific points:

Figure 1A: pACC seems activated in lane 4 (just with IFN γ /TNF α) treatment... this is shown in the quantification as well - thus it seems like cytokine treatment is sufficient to activate AMPK, which would argue against the entire thesis of the paper (that AMPK activation blocks cytokine effects). There's no real difference in pACC with AICAR above IFN γ /TNF α . The authors make note of this, but don't seem to recognize that this undercuts the entire paper - what's the point of adding an AMPK activator to counter cytokine induced atrophy, if the cytokines are already activating AMPK, and causing atrophy? (Same point re pAMPK)”.

To address this point, we assessed the activation time course of AMPK in the context of cytokine treatment with and without AICAR or metformin. We observed that the AMPK agonists activate AMPK at a time before it is activated by the cytokines alone (Fig. S1). Therefore, we are inducing AMPK activation before it is normally activated by inflammation. However, we still observe that metformin is ineffective, whereas AICAR is effective at preventing atrophy. As detailed in our manuscript, this discrepancy likely arises from the differential effects on metabolic function. Our results suggest that activation of AMPK by AICAR before the induction of mitochondrial dysfunction appears to be able to protect muscle from wasting. However, activation of AMPK through the induction of mitochondrial dysfunction, as is the case for metformin and likely the case for cytokine-treatment/inflammation, likely contributes to inhibition of anabolism.

“**Figure 3:** the relative pS6/S6 determination is a bit misleading, given that the cytokines are dramatically decreasing S6 levels. Also, there is an inconsistency in total S6 across the lanes, making this experiment difficult to interpret. This should be repeated, along with phospho-p70S6K/total p70 determination, along with the whole blot being shown”.

We apologize for the inconsistent loading in the presented western. We have now provided an alternative experiment which addresses this issue. We have also included blots for phospho-S6K throughout our manuscript to corroborate our findings with phospho-S6.

“**Figure 4:** Decreasing total levels of iNOS was not the prior mechanism leading to atrophy - rather, releasing iNOS from the dystroglycan complex, leading to decreased Akt phosphorylation. It would be surprising if AMPK was perturbing that mechanism... so how do the authors think iNOS is functioning here?”

We and others have previously shown that iNOS is a key contributor to muscle wasting (PMID: 21832306; PMID: 28264935). It has previously been shown that iNOS expression mediates the downregulation of MyoD mRNA (PMID: 16024790). It has also been shown to suppress Jun-D activity and protein synthesis (PMID: 8617220, 19470832, 19295495). A full understanding of the mechanism of how iNOS contributes to muscle wasting has yet to be conclusively demonstrated and is currently under investigation but is outside the scope of this manuscript. Nevertheless, its importance as a pro-cachectic factor is established, and so assessment of its expression provides further understanding of how AICAR, but not metformin, can prevent cytokine-induced muscle wasting.

“**Figure 7:** What is the effect of AICAR alone on muscle mass? Here too, examination of the E3 ligases would seem to be essential. (MuRF1 and MAFbx/Atrogin-1)”

We have assessed muscle mass and MuRF1 and MAFbx/Atrogin-1 in control mice for both AICAR and metformin for both our C26 and LPS models (see above) (Fig. S5, S8). There was no significant effect of these compounds on muscle mass when treated alone in either mouse line. In addition there was no significant effects on E3-ligase expression. There was a trend towards decreased E3-ligase expression in the metformin treated control for the LPS model. However, this effect was not observed in the LPS + metformin treated cohort, nor in the metformin treated cohorts in the C26 model. Therefore, we do not think this trend is biologically significant.

Referee #3 (Comments on Novelty/Model System for Author):

“C2C12 myotubes treated with IFN γ /TNF α are well accepted in vitro model of cytokine induced muscle wasting; intra-muscular injection of IFN γ /TNF α cytokines in mice provides a tumor-free in vivo model of muscle wasting; C26 adenocarcinoma tumor-bearing mice are in vivo cachectic model of muscle wasting”.

Referee #3 (Remarks for Author):

The paper of Hall et al is an interesting and elegant study, which identifies a novel protective function of AMPK against cytokine-driven atrophy. By using in vitro and in vivo models of cytokine-driven muscle wasting, the authors have shown that the AMPK agonist AICAR suppresses cytokine-induced atrophy. Prevention of atrophy was associated with reduction of glycolytic flux, restoration of oxidative metabolism and suppression of iNOS/NO pathway in the context of IFN γ /TNF α treatment. Compelling evidence indicates that the protective role of AICAR is mediated by AMPK and is linked to its anti-inflammatory properties, being independent from its well-known role in suppression of anabolism upon metabolic stress. Indeed, the reduction of mTOR signaling and protein synthesis, induced by IFN γ /TNF α , was blunted by AICAR concomitant stimulation. In line with this, the ability of AICAR treatment to restore mTOR signaling was impaired in the presence of Compound C, a specific AMPK inhibitor. This is a phenomenon that deserves further exploration and the present report contributes to that effort. The manuscript is technically sound and the results are very convincing. The experiments are carefully performed and the results are clearly presented. I would recommend publication of this paper essentially as it is.

Minor observations:

Line 118: "Fig. 1A"

Lines 131-132: rephrase "AICAR treatment, but not metformin trended to higher mRNA levels for MyoD (significant) and myogenin"

Fig. 1A: the right panel lacks of p values on the bars

Line 317: "... sufficient to induce cachexia"

We thank the reviewer for his corrections and support of our work. All observations have been corrected in the text.

2nd Editorial Decision

16 April 2018

Thank you for the submission of your revised manuscript to EMBO Molecular Medicine. We have now received the enclosed reports from the referees that were asked to re-assess it. Please accept our sincere apologies for the delay, as unfortunately, one referee couldn't help any longer. As you will see the reviewer who assessed the revised paper is now supportive and I am pleased to inform you that we will be able to accept your manuscript pending final editorial amendments.

Please submit your revised manuscript within two weeks. I look forward to seeing a revised

form of your manuscript as soon as possible.

I look forward to reading a new revised version of your manuscript as soon as possible.

**** Reviewer's comments ****

Referee #1 (Remarks for Author):

The authors have responded admirably to the critiques and a stronger, more important study has emerged. The sepsis results are particularly interesting and I look forward to further developments.

2nd Revision - authors' response

01 May 2018

Authors made the requested editorial changes.

Corresponding Author Name: Dr. Imed Gallouzi
Journal Submitted to: EMBO Molecular Medicine
Manuscript Number: 2017-08307